# Bioassessment of Macroinvertebrate Communities Influenced by Gradients of Human Activities

**DOI:** 10.3390/insects15020131

**Published:** 2024-02-14

**Authors:** Rui Li, Xianfu Li, Ronglong Yang, Muhammad Farooq, Zhen Tian, Yaning Xu, Nan Shao, Shuoran Liu, Wen Xiao

**Affiliations:** 1Institute of Eastern-Himalaya Biodiversity Research, Dali University, Dali 671003, China; 18849641952@163.com (R.L.); lixf@eastern-himalaya.cn (X.L.); rlyang_7490@163.com (R.Y.); farooq@eastern-himalaya.cn (M.F.); yaningxu0905@163.com (Y.X.); shao20010811@163.com (N.S.); 2Collaborative Innovation Center for Biodiversity and Conservation in the Three Parallel Rivers Region of China, Dali 671003, China; 3The Provincial Innovation Team of Biodiversity Conservation and Utility of the Three Parallel Rivers Region, Dali University, Dali 671003, China; 4State Key Laboratory of Freshwater Ecology and Biotechnology, Institute of Hydrobiology, Chinese Academy of Sciences, Wuhan 430072, China; tianzhen@ihb.ac.cn; 5Yunling Black-and-White Snub-Nosed Monkey Observation and Research Station of Yunnan Province, Dali 671003, China

**Keywords:** macroinvertebrate, Cangshan streams, upstream and downstream habitats, watershed topographic segmentation, homogenization

## Abstract

**Simple Summary:**

Watersheds, as carriers of biodiversity with distinct boundaries, can provide the basic conditions for the distribution and dispersal of species. This study found that frequent and complex land-use type changes caused by human activities can alter river conditions, thus affecting aquatic biodiversity as well as dispersal and distribution. The disturbed streams had lower aquatic biodiversity than those in their natural state, a decrease in disturbance-sensitive aquatic insect taxa and a more similar community structure. In natural woodland areas, species distributions may be constrained by watershed segmentation and may present more complex community characteristics.

**Abstract:**

This study explores the impact of anthropogenic land use changes on the macroinvertebrate community structure in the streams of the Cangshan Mountains. Through field collections of macroinvertebrates, measurement of water environments, and delineation of riparian zone land use in eight streams, we analyzed the relationship between land use types, stream water environments, and macroinvertebrate diversities. The results demonstrate urban land use type and water temperature are the key environmental factors driving the differences in macroinvertebrate communities up-, mid-, and downstream. The disturbed streams had lower aquatic biodiversity than those in their natural state, showing a decrease in disturbance-sensitive aquatic insect taxa and a more similar community structure. In the natural woodland area, species distributions may be constrained by watershed segmentation and present more complex community characteristics.

## 1. Introduction

In recent years, there has been increasing concern about the health of aquatic ecosystems, especially those near urban areas [1,2]. This concern is not only related to human production [3], daily life, and social development but also, more importantly, reflects the coordination between urban development and ecosystem protection through the indirect reflection of aquatic biodiversity [4,5,6]. Urban development is the result of human activities and brings about a series of environmental issues [7,8]. Among the systems damaged, aquatic ecosystems are some of the most challenging to restore, highlighting the contradiction between development and protection [9,10].

Urban development relies heavily on the support of aquatic ecosystems [11]. However, human activities during the urbanization process inevitably affect aquatic ecosystems to varying degrees [12]. Previous research has mainly focused on biodiversity loss caused by pollutant emissions and changes in community structure due to hydraulic facilities [13,14,15]. However, the issue of the resulting dispersal distribution of species has been neglected and may be the root cause of community change in aquatic ecosystems disturbed by human activities [16]. Therefore, it is particularly important to use the biological index as an evaluation of river health. Water quality biological evaluation refers to the evaluation of the biological quality of water bodies through the investigation or direct detection of aquatic organisms in water bodies. The Family Biological Index (FBI) was proposed by the American scholar Hilsenhoff [17] in 1988. In order to reduce the difficulty of species identification, save time and realize the rapid evaluation of river health, he proposed the FBI index on the basis of the Hilsenhoff Biological Index (HBI), established by him, which effectively promoted the wide application of the Biological Index (BI).

Macroinvertebrates are aquatic organisms that spend all or part of their life history on the bottom surface or substrate of a body of freshwater (both flowing and standing) and whose individuals are unable to pass through a 425 μm (40 mesh) mesh screen. Aquatic insects are an important part of the macroinvertebrate community, most of whose species are able to cross both land and water interfaces and exhibit flight characteristics [18,19,20]. Depending on the strength of the adults’ migratory ability and the larvae’s drifting habits downstream, their community structure may show patterns related to different habitat conditions. These patterns depend on the ecological niches of the biological groups themselves [21,22,23]. In mountainous systems, these patterns are also related to the elevation gradient [24,25]. Elevation changes regulate the distribution of vegetation zones, determining the local environmental conditions along the streams and their coupled relationships with vegetation zones [26,27,28,29]. Therefore, species’ ecological niches and terrestrial ecosystems jointly determine the community structure of macroinvertebrates, and this relationship can be observed within large-scale elevation ranges [30,31].

Stream ecosystems near urban areas are often altered by human activities, changing the terrestrial ecosystems along riparian zones and the surface runoff hydrology of streams [32,33]. Consequently, environmental factors of the water bodies and the structure of biological communities are affected [34]. This is mainly reflected in urban stream ecosystems where the channels have been altered due to irrigation and landscaping needs, leading to changes such as altering water flow directions, channel bottom hardening, and nutrient input from fertilizer residues [35,36,37]. As a result, previously isolated channels might intersect through tributary networks, allowing water exchange between branches, and affecting downstream water quality [38,39]. Altered connectivity, in turn, may impact macroinvertebrate pathways for dispersal and the distribution of communities might change accordingly [40,41,42]. Increased connectivity among various habitat networks complicates the study of biodiversity patterns in macroinvertebrates [43,44].

There are various methods of dispersal known among macroinvertebrates in streams [45]. The most common one is longitudinal dispersal along the upstream and downstream flow of the stream, where larvae disperse downstream during their juvenile stage and adults migrate upstream during their adult stage [46,47]. However, latitudinal dispersal can also occur. In mountainous regions, especially in streams separated by ridges, the latitudinal dispersal of aquatic insects might be limited by topographical factors and the insects’ flight capabilities [48]. However, most studies of these systems have focused on linear network branches, ignoring the spatial relationships between branches in the broader network and the connectivity within and between branches [49,50]. It is therefore critical to determine how aquatic biodiversity and community structure in alpine streams respond to anthropogenic changes.

To explain the dispersal pathways of macroinvertebrates among streams and the driving factors behind community formation, we selected parallel streams as the study objects. The upstream sections were natural habitats with ridges as geographical barriers, while the downstream sections were influenced by human activities, with flat terrain and confluences of tributaries. We propose the following hypotheses: (1) in the headwaters, the geographic isolation that exists between streams can impede the dispersal of environmental factors and aquatic communities, thus allowing them to exhibit heterogeneity; (2) in the middle and lower reaches, anthropogenic disturbances may alter stream connectivity, resulting in more even dispersal between biotic and abiotic organisms and, thus, homogeneity of the stream water environment and aquatic communities. Therefore, this study provides a scientific basis for sustainable development in watershed management, biodiversity conservation, and the functions of ecosystem services.

## 2. Material and Methods

### 2.1. Land Use Classification and Sites

The study area is located in Cangshan Erhai National Nature Reserve, northwest Yunnan, China. Among the eighteen streams in Cangshan Mountain, eight streams were chosen as study sites: Wanhua Creek, Mangyong Creek, Jin Creek, Baishi Creek, Shuangyuang Creek, Yinxian Creek, Zhonghe Creek, and Baihe Creek. For each of these streams, three sample sites were designated, representing the up-, mid-, and downstream sections. Each sample site had five replicates. In total, 23 sample sites were selected for the collection of macroinvertebrate fauna and the assessment of aquatic environmental factors in November 2019. Of these, the sample site at the upstream end of Zhonghe Creek (ZHX-1) was not collected because the terrain was too steep to be reached by humans (Figure 1 and Figure 2).

Sample sites in the upper reaches of the study area are largely free of significant human activity and can be considered natural woodland environments with the presence of ridges between streams. For species with the potential to migrate over long distances, the presence of ridges resulted in low dispersal rates, which impeded inter-population gene flow, and may lead to high levels of differentiation in mountain macroinvertebrate populations. The middle and lower reaches are mostly inhabited areas, with large amounts of building land and agricultural land. Therefore, Google Earth Pro (version 7.3.2.5776) software was used to manually circle and classify the types of land use within the watershed. Additionally, the areas of each land use type were measured using the software, resulting in three primary types: Forest Land (Forest Land), Crop Land (Cropland), and Urban Land (Urban Land). Subsequently, Arc Map 10.8 was utilized to combine the sample point map with the watershed land use area map, and distinct color codes were employed to differentiate between these categories (Figure 2). Area of three land use types in each basin (Table 1).

### 2.2. Biological Data Collection and Identification

#### 2.2.1. Macroinvertebrate Sample Collection and Identification

Macroinvertebrates were collected using a Surber net with a pore size of 0.5 mm, featuring a sampling area of 0.09 m^2^, positioned at the riverbed bottom and aligned with the water flow direction. Five replicates were collected at each site. The collection process consisted of hand sweeping macroinvertebrates on the stone surface into the net and then gently hand-stirring the substrate for the macroinvertebrates to enter the net with the current. After this initial phase of coarse filtration, the collected specimens were carefully transferred into specimen bottles and preserved in 95% ethanol. Additionally, details about habitat conditions, such as the type of substrate and the presence of leaf litter, were recorded at each sample site. The majority of macroinvertebrates were assigned to the genus or species level, with Chironomidae classified at the family or subfamily level, as per references [51,52,53,54]. Furthermore, the macroinvertebrate specimens underwent verification by the authors responsible for the primary identifications.

#### 2.2.2. Determination of Environmental Factors

At the sampling sites, various water quality parameters were measured using a portable water quality analyzer (ProPlus, YSI, Yellow Springs, OH, USA). These parameters included water temperature (WT), conductivity (Cond), pH, dissolved oxygen (DO), oxidation–reduction potential (ORP), total dissolved solids (TDS), and salinity. Additionally, river width (width) and flow velocity (FV) were assessed using a direct-reading flow velocity meter, specifically the FP211 model from the USA. The chlorophyll-a (Chl-a) content within the water column was quantified spectrophotometrically in the laboratory, following established measurement protocols.

**Figure 2 insects-15-00131-f002:**
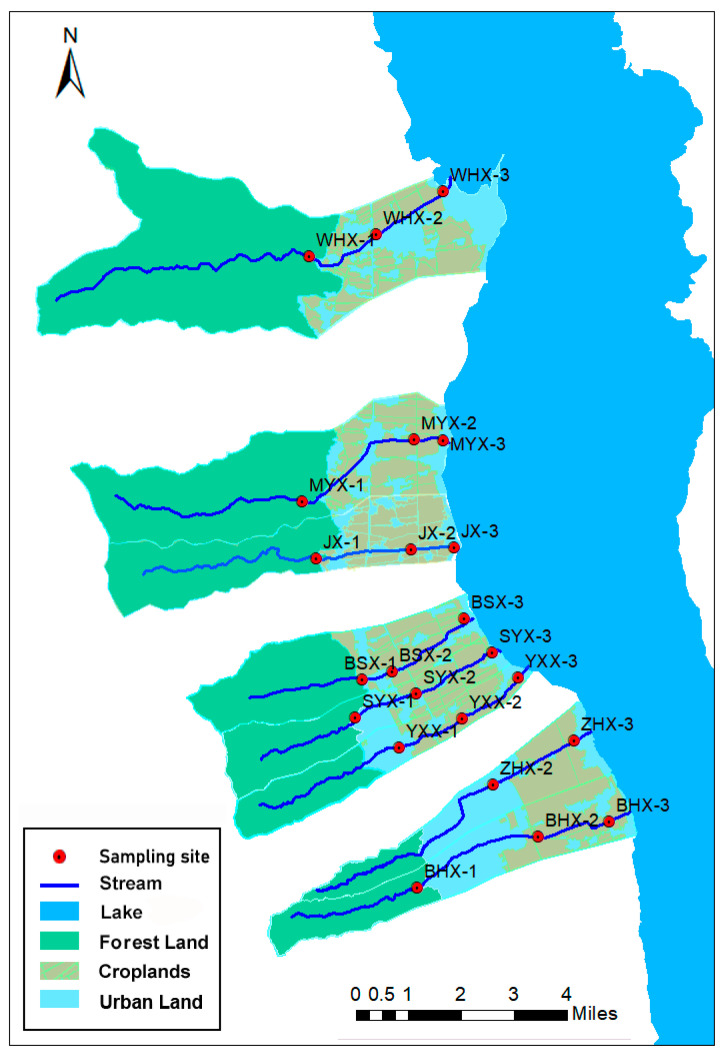
Land use type division map. The land use types were categorized into three basic types: forest land (Forest Land), cropland (Cropland), and building land (Urban Land).

### 2.3. Data Processing

#### 2.3.1. Environmental Factors

Upstream and midstream environmental data were analyzed for differences in SPSS software (version 25.0). First, the data were assessed for normal distribution using the Kolmogorov–Smirnov test (K–S test). If the data followed a normal distribution, a one-way ANOVA was employed. In cases where the data didn’t conform to a normal distribution, the non-parametric Kruskal–Wallis test was utilized. To ascertain the environmental heterogeneity between each site, an unconstrained linear model’s principal component analysis (PCA) was conducted using the R programming language.

#### 2.3.2. Macroinvertebrate Diversity Analysis

Community alpha diversity was described using several metrics: Species Richness, dominant species, Shannon–Wiener index, and Pielou index. Species variables underwent log-transformation to achieve approximate normality, while environmental variables were Z-standardized using SPSS 25.0. To assess the variability of alpha diversity in the upper and middle reaches, diversity indices were subjected to normality testing. A one-way ANOVA was used for indices conforming to a normal distribution, and the Kruskal–Wallis nonparametric test was employed for those not conforming. The Spearman’s rank correlation analysis was used to determine the significance of the relationship between the macroinvertebrate diversity indices and the proportional areas of land use types estimated on the basis of data in Table 1.

The detailed calculation process for assessing the pollution-resistant taxa of macroinvertebrates using the FBI index is:$$I_{\mathrm{FBI}}=\sum_{i=1}^{N}\frac{g_{i}m_{i}}{N}$$
where *N* is the total number of macroinvertebrate individuals in the sample; *g_i_* is the number of individuals in the *i* th section-level taxon of macroinvertebrates; *m_i_* is the stain resistance value of section-level taxon *i*.

Before selecting the appropriate means of correlation analysis, we performed a detrended correspondence analysis (DCA) on the biome data and determined whether the community structure was a unimodal or linear model. The DCA analysis showed that the FLG value (Final Length of Gradient, FLG) was 2.017.

Screening of environmental factors: Collinearity analysis was performed for all environmental factors. The largest variable is deleted in turn, namely the environmental factor of collinearity, until all the variables are less than 10. Then, we detected the lowest AIC (Akaike Information Criterion) value using a step model, in which the model automatically filtered out the best environmental factors. The environmental factors we included in the RDA analysis were: water temperature (WT), conductivity (Cond), dissolved oxygen (DO), oxidation–reduction potential (ORP), flow velocity (FV) and urban land (UL) (R 4.3.1).

## 3. Results

### 3.1. Physical and Chemical Characteristics of Water Bodies

Water chemistry indicators reflecting the trophic state and physical characteristics of the water bodies exhibited no significant differences (*p* > 0.05) between the upper and middle reaches. However, the average values of parameters such as Cond, Sal, TDS, pH, Chl-a, WT, and FV displayed a gradual increase from the upper to the lower reaches. The disparities in the proportions of FL, CL, and UL had significant differences (*p* < 0.01) between the upper and middle reaches (as presented in Table 2).

The Principal Component Analysis (PCA) ranking of environmental factors for each sample site (depicted in Figure 3) indicated that polygons representing the PCA rankings for the upstream, midstream, and downstream regions did not overlap, signifying substantial environmental heterogeneity among these sections. A comparison of upstream habitats with downstream ones showed that the minimum convex polygons encompassing downstream habitats had a larger area, suggesting greater environmental heterogeneity among the downstream sites. However, adjacent sample sites exhibited a high degree of environmental similarity. The areas of the midstream and downstream polygons intersected, indicating greater environmental similarity between these areas. FL and ORP at upstream sites are the main influencing factors affecting environmental heterogeneity in upstream and other reaches. The flow rate only affects site BSX-3 (Figure 3).

**Figure 3 insects-15-00131-f003:**
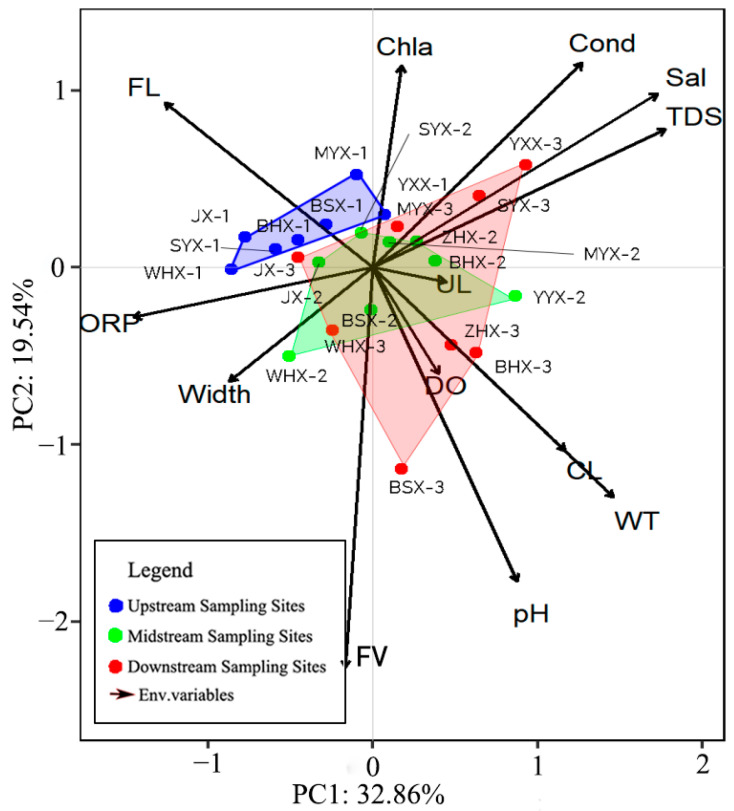
Principal Component Analysis (PCA) plots for upstream, midstream, and downstream related environmental variables. (WT: water temperature, Cond: conductivity, DO: dissolved oxygen, ORP: oxidation–reduction potential, TDS: total dissolved solids, Sal: salinity, Width: stream width, FV: flow velocity, Chla: chlorophyll-a, FL: Forest Land, UL: Urban Land, CL: Crop Land.).

### 3.2. Macroinvertebrate Community Structure

#### 3.2.1. Differences in Species Composition

The mean FBI index was less than the middle and downstream ones (Table 3). The table of relative abundance of species composition of macroinvertebrates in the upstream, midstream, and downstream regions shows that Plecoptera occurs only upstream and that *Kamimuria*, *Cryptoterla* sp1., and *Amphinemura* sp1. have a low fouling tolerance value. The species composition upstream is generally that of species with low fouling tolerance values (Appendix A).

Through SIMPER analysis [55], several species with relatively different contributions to the upstream, midstream, and downstream community aggregation were identified, and the species causing the difference between the upstream and downstream communities were *Baetis* sp1., *Gammarus* sp1., *Chironominae*, *Orthocladiiae*, *Limnodrilus* sp1., and *Baetis* sp2. Species causing differences in the upstream and downstream communities are *Baetis* sp1., *Orthocladiiae*, *Gammarus* sp1., *Limnodrilus* sp1., *Baetis* sp2., *Chironominae.* Species that cause differences in the midstream and downstream communities are *Baetis* sp1., *Gammarus* sp1., *Orthocladiiae*, *Limnodrilus* sp1., *Chironominae* (Table 4).

#### 3.2.2. Macroinvertebrate Species Diversity

Differential analysis of the diversity index of the upper, middle, and lower macroinvertebrates revealed no significant differences (*p* > 0.05). However, the upstream diversity index was larger than the middle and downstream ones: Richness indices showed the trend of upstream > middle > downstream; Shannon–Wiener index and Pielou showed the upstream > downstream > middle trend (Figure 4). This trend suggests that the upper reaches are species-rich hotspots, while the middle and lower reaches have relatively low macroinvertebrate abundance and simple species composition. Notably, Species Richness displayed positive correlations with woodland land use, while the Shannon–Wiener index showcased a negative correlation with urban land use (Table 5).

### 3.3. Relationship among Macroinvertebrates and Basin-Scale Environmental Factors

The macroinvertebrate communities of the upstream, midstream, and downstream sections exhibited partial overlap on the RDA ordination map (Figure 5). Urban Land use type and water temperature is the main environmental factor causing the differences in macroinvertebrate communities upstream, midstream, and downstream. Upstream, macroinvertebrate community differences are mainly driven by DO, and the differences in middle and lower communities are mainly affected by ORP, Cond, and water flow velocity.

## 4. Discussion

### 4.1. Effect of Human Activities on the Characteristics of the Water Environment

The analysis of the physicochemical properties of water in three types of ecological sites revealed a gradual increase in the average values of parameters such as Cond, Sal, TDS, pH, Chl-a, WT, and FV from the upstream to the downstream (Table 2). The areas of agricultural land and urban construction land in the middle and lower reaches noticeably increased compared to the upstream ones (Table 2). This indicates that the middle and lower reaches have undergone significant changes in the water environment due to human activities, as depicted in Figure 3. This result is consistent with common features observed in studies on the degradation of the ecological quality of urban streams [56,57].

Possible reasons for these results are that in the middle and lower reaches, rivers pass through urban residential areas and agricultural buffer zones before entering lakes. These areas are densely populated, with high levels of anthropogenic disturbance. To meet the demands of production and daily life, rivers are filled, cut, and hardened, resulting in a reduction in the water area of rivers and a network-like or branching structure. With the sharp decrease in forested areas and an increase in areas covered by roads, buildings, and agricultural land [58], erosion of embankments and riverbeds intensifies, accelerating the influx of nutrients and sediments into the river channel within the watershed. This severely impacts the morphology of the river channel and the quality of the habitat [59].

Despite the vertical connectivity of water bodies in the upper, middle, and lower reaches observed in this study, the upstream maintains unique habitat types, low temperatures, and highly permeable mineral substrates [60]. The proportion of forested areas in the middle and lower reaches is smaller, with reduced vegetation cover along river corridors. Direct sunlight exposure leads to elevated water temperature and increased flow velocity. The predominant substrate type is impermeable artificial cement, resulting in accelerated flow rates [61]. Additionally, channelization of the river increases the rate of material transport and exchange, enhancing the spread of pollutants, thereby significantly reducing the complexity of river habitats and gradually leading to homogenization of the watershed water environment [62].

### 4.2. Effects of Watershed Isolation on Water Environment Characteristics and Biological Communities

Spatial and environmental factors are crucial driving forces for community structure, including factors such as species dispersal ability, landscape configuration, and spatial distribution patterns of interacting species [63]. The Cangshan Creek system originates from mountainous vegetation, with upstream streams arranged in a parallel pattern, exhibiting distinct geographical barriers. The upstream site, inside the canyon, is blocked off horizontally by a ridge, hindering the spread of flying species. In general, lower hydrological connectivity can impact species dispersal, leading to a few species becoming dominant and thereby suppressing overall diversity of macroinvertebrates [64]. However, the upstream areas are mostly covered by forests, and the forested areas promote and protect macroinvertebrate diversity through mechanisms such as pollutant interception, providing organic matter like logs and leaf litter, enhancing shade in the river, and lowering water temperature [36]. This, in turn, contributes to the stability and integrity of river food chains and webs [65,66]. Due to the high forest cover in the upstream regions, although micro-scale environmental heterogeneity is lower than in the middle and lower reaches, the complexity of community structure remains significantly higher than in the middle and lower reaches.

### 4.3. Effects of Water Environment Characteristics and Connectivity on Biological Dispersal

The structure of macroinvertebrate communities is influenced by the characteristics of the habitats they inhabit, including physical and water chemistry aspects. These habitat features, in turn, are influenced by both natural factors at the watershed scale and anthropogenic disturbances [67]. While species dispersal in space promotes species movement and colonization in new locations, environmental selection can still alter community composition [68]. In the middle and lower reaches of the stream, human activities have influenced changes in water environmental factors. As a result, this has led to a decrease in environmentally sensitive taxa and an increase in pollution-tolerant taxa of macroinvertebrates [69].

Regarding species composition, despite maintaining a certain degree of connectivity between the upstream, midstream, and downstream areas, the changes in water quality caused by pollutants discharged in the middle and lower reaches are particularly detrimental to sensitive species [70]. Over long-term adaptation, these communities show a reduction or loss of sensitive taxa and an increase in pollution-tolerant taxa, resulting in homogenization of macroinvertebrate communities. This phenomenon is evident in the decrease in the number of macroinvertebrate species from 58 in the upstream to 45–46 in the middle and lower reaches. The upstream has rich unique taxa and sensitive species, such as Plecoptera, while the dominant groups in the middle and lower reaches are mainly composed of species with higher tolerance to pollution, such as Gammarus and Tubificidae. Additionally, human activities have enhanced the latitudinal connectivity of river channels, further contributing to the homogenization of macroinvertebrate species composition. Wang, X. and Tan, X. also demonstrated this in their study [71]. The research indicates that changes in land use types caused by human activities lead to homogenization in the species composition of macroinvertebrates, supporting the second hypothesis proposed in this paper.

In terms of species diversitys species richness shows a positive correlation with forest land use types, while the Shannon–Wiener index is negatively correlated with urban land use types (Table 5). These findings are consistent with previous research results [72,73]. Human activities, including channelization, straightening, and consolidation in the middle and lower reaches of rivers, lead to the loss of natural stream meandering, changes in habitat topography, and alterations in hydrological conditions. As a result, suitable habitats for macroinvertebrates decrease, and the community of macroinvertebrates is influenced by pesticide use, fertilizer application, and direct pollutant discharge (Figure 5). The research indicates that differences in land use types are a key driving factor for the decrease in diversity of macroinvertebrates [74,75].

## 5. Conclusions

The main species responsible for the difference in species composition between upper, middle, and downstream regions were *Baetis* sp1., *Gammarus* sp1., *Chironominae*, *Orthocladiiae*, *Limnodrilus* sp1., and *Baetis* sp2. The mean FBI index was less than the middle and downstream ones. The species composition upstream was generally that of species with low fouling tolerance values (Appendix A). The Plecoptera found only upstream, *Kamimuria*, *Cryptoterla* sp1., and *Amphinemura* sp1., have low fouling tolerance values.

The disturbed streams had lower aquatic biodiversity than those in their natural state, a decrease in disturbance-sensitive aquatic insect taxa, and a more similar community structure. In the natural woodland area, species distributions may be constrained by watershed segmentation and present more complex community characteristics. Conversely, heavily impacted areas witness the depletion of forested and grassland water zones due to urban expansion and the fragmentation of watershed landscapes. Urban land use type and water temperature are the main environmental factors causing the differences in macroinvertebrate communities upstream, midstream, and downstream.

## Figures and Tables

**Figure 1 insects-15-00131-f001:**
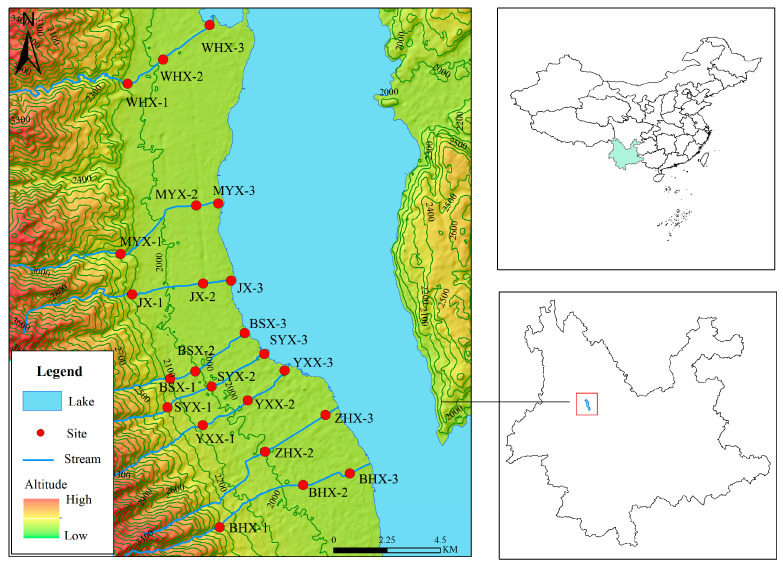
Distribution of sampling sites in the Cangshan Mountain region. The sampling sites are located in northwest of Yunnan province, China, with an elevation range of 1959–2282 m. Annotation: 1, 2, and 3 represent upstream, midstream, and downstream respectively. WHX: Wanhua Creek, MYX: Mangyong Creek, JX: Jin Creek, BSX: Baishi Creek, SYX: Shuangyuang Creek, YXX: Yinxian Creek, ZHX: Zhonghe Creek, BHX: Baihe Creek.

**Figure 4 insects-15-00131-f004:**
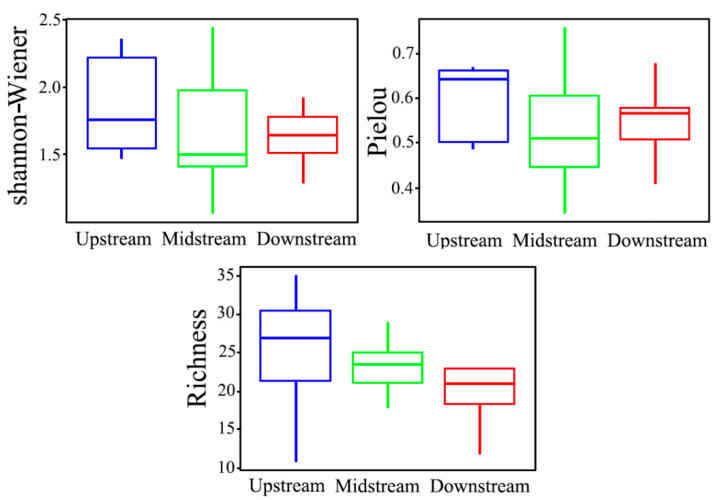
Macroinvertebrate diversity index.

**Figure 5 insects-15-00131-f005:**
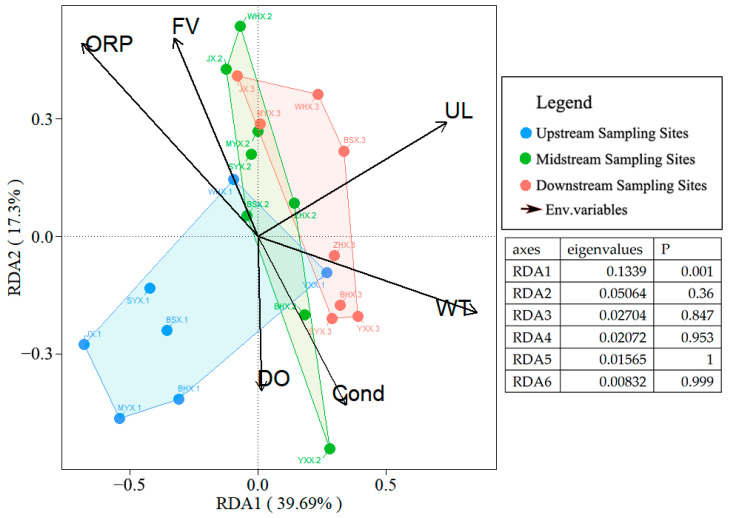
Biplot of RDA calculated based on the macroinvertebrate community data and environmental factors. Eigenvalues values and the *p*-values of the axes. The total *p*-value of the RDA model was 0.028, (WT: water temperature, Cond: conductivity, DO: dissolved oxygen, ORP: oxidation–reduction potential, FV: flow velocity, UL: Urban Land).

**Table 1 insects-15-00131-t001:** Cangshan stream river basin area.

Site	Forest Land (m^2^)	Cropland (m^2^)	Urban Land (m^2^)	Total
BHX	5,680,987	5,029,708	5,915,036	16,625,731
ZHX	3,013,559	3,994,146	5,623,893	12,631,598
YXX	5,622,397	2,707,984	2,841,034	11,171,415
SYX	6,838,849	2,919,541	2,692,674	12,451,064
BSX	9,783,803	4,551,497	1,913,736	16,249,036
MYX	18,119,389	6,399,511	4,187,368	28,706,268
JX	12,933,323	3,809,709	3,506,424	20,249,456
WHX	32,805,788	7,721,723	8,697,835	49,225,346

**Table 2 insects-15-00131-t002:** Description and difference analysis of environmental factors in upper, middle, and lower reaches. A one-way analysis of variance (superscript 1) and a nonparametric test (superscript 2) were used to compare the difference of environmental factors between the upper and middle reaches. ** refers to *p* < 0.01.

Habitat Factor		Upstream			Midstream			Downstream		
		Min	Max	Mean	Min	Max	Mean	Min	Max	Mean
Chemical factor	Cond (μs/cm) ^1^	41.90	193.00	107.33	54.30	163.00	119.98	63.10	220.30	121.36
	Sal (ng/L) ^1^	0.03	0.11	0.06	0.03	0.14	0.09	0.04	0.22	0.10
	TDS (g/L) ^1^	37.70	148.85	76.79	42.90	243.00	127.47	6.75	299.00	129.22
	ORP (mV) ^1^	24.10	73.40	56.16	20.70	80.70	53.44	−18.60	80.70	38.28
	DO (mg/L) ^2^	4.13	8.41	7.13	4.93	13.57	7.64	5.12	11.86	7.46
	pH ^2^	6.91	7.90	7.50	7.25	8.81	7.80	7.10	9.77	8.14
	Chl-a (mg/cm^2^) ^2^	0.38	5.88	2.52	0.56	6.49	3.12	0.21	14.03	5.08
Physical factor	WT (°C) ^1^	9.40	19.30	13.90	13.40	21.50	16.71	12.20	22.50	17.44
	Width (m) ^1^	1.40	7.00	3.69	1.00	8.00	4.89	0.30	7.00	3.76
	FV (m/s) ^1^	0.00	0.26	0.13	0.00	0.58	0.18	0.04	0.84	0.25
Land-use type	Forest Land (%)^2^ **	81.20	100.00	95.07	0.00	25.60	9.64	0.00	0.00	0.00
	Cropland (%)^2^ **	0.00	6.90	2.33	0.00	80.90	44.98	68.80	89.60	80.26
	Urban Land (%)^2^ **	0.00	11.90	2.60	15.40	100.00	45.38	10.40	31.20	19.74

**Table 3 insects-15-00131-t003:** FBI (Family biotic index) at various points of Cangshan streams.

Site	Upstream	Midstream	Downstream
BHX	4.3	5.7	6.2
ZHX	-	5.6	5.5
YXX	5.8	5.5	7.7
SYX	3.6	3.6	7.7
BSX	4.0	3.9	5.3
JX	3.9	5.5	5.3
MYX	4.2	6.6	6.8
WHX	5.8	5.6	5.9
Average value	4.6	5.1	6.4

**Table 4 insects-15-00131-t004:** A SIMPER analysis of differences in aggregate contributions of upstream, midstream, and downstream communities (1: upstream, 2: midstream, 3: downstream. Contribution of species to Bray–Curtis phase divergence between mean groups).

	Species Numbering	Cusum	Average
$’1_2’	sp35	0.2244153	0.14642
	sp68	0.4046277	0.11758
	sp28	0.501261	0.06305
	sp26	0.5976117	0.06286
	sp58	0.6563657	0.03833
	sp36	0.7009538	0.02909
$’1_3’	sp35	0.2693694	0.20681
	sp26	0.4047368	0.10393
	sp68	0.5180737	0.08702
	sp58	0.6188757	0.07739
	sp36	0.6768917	0.04454
	sp28	0.7179495	0.03152
$’2_3’	sp35	0.2565883	0.19059
	sp68	0.4332157	0.1312
	sp26	0.562946	0.09636
	sp58	0.6554321	0.0687

**Table 5 insects-15-00131-t005:** Spearman correlation coefficient of land use type and macroinvertebrate diversity index. The proportion of land use types to correspondent total watershed area was used for analysis.

	Shannon–Wiener	Richness	Pielou	Forest Land	Cropland
Shannon.wiener					
Richness	0.582 **				
Pielou	0.839 **	0.162			
Forest Land	0.364	0.530 **	0.197		
Cropland	−0.037	−0.321	0.111	−0.777 **	
Urban Land	−0.448 *	−0.287	−0.396	−0.605 **	0.236

* indicates a significant (two-tailed) correlation at the 0.05 level; ** indicates a significant (two-tailed) correlation at the 0.01 level. In each case N = 23. The proportional areas of land use types were derived from data in Table 1.

## Data Availability

Data is contained within the article or Appendix A.

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
