# Peer review of "Bioassessment of Macroinvertebrate Communities Influenced by Gradients of Human Activities"

_insects, 2024, doi:10.3390/insects15020131_

Round 1
Reviewer 1 Report
Comments and Suggestions for Authors
Overall Comments
The manuscript contributes a relevant piece to a large existing body of literature on the impacts of human-induced disturbance on aquatic macroinvertebrates. The study includes a well-chosen suite of environmental and diversity response variables for analysis, and details regarding the study’s sampling methodology are reasonably well-explained. References to some of the relevant disturbance literature are included, and overall the manuscript is reasonably well-written. However, there are some notable concerns that must be addressed in three general areas. First, a more thorough discussion of the disturbance literature is needed to more fully explain the effects of human-induced disturbance on diversity. Intermediate levels of disturbance are known to sometimes aid diversity by preventing dominance, such as discussed in the Roxburgh et al. article you cite. However, this is not sufficiently addressed in the present manuscript and you appear to argue for a strict linear relationship between increasing disturbance and declining diversity. Also, the claim about most precise level of taxonomy does not match the data reported in Fig. 3, which only reports macroinvertebrate captures at the order level. Additionally, some minor formatting and grammatical errors must be corrected. Please see the specific comments and suggestions below.
Specific Comments
Line 26 – A space is needed after the colon and “This” does not need to be in bold.
Lines 60-62 – This is presently an incomplete sentence. Also, the statement incorrectly lumps flighted adult insects in with benthic macroinvertebrates. Please note that following metamorphosis to the flighted adult stage, many formerly aquatic macroinvertebrates are no longer in a life stage that can be termed “benthic.” Please amend the wording to resolve both concerns.
Line 69 – Because of the usage here, “species” needs an apostrophe after it.
Line 80 – The phrase “in terms of biology” is superfluous and can be eliminated.
Line 85 – The phrase “various ways for the dispersal” is awkward. I recommend changing this to “various methods of dispersal known among…”
Lines 90-91 – As this sentence is currently worded it is not clear whether “their own” refers to “aquatic insects” or “topographical factors.” It is presumably the former, but the present phrasing leaves this ambiguous. I recommend changing this to say “…and the insects’ flight capabilities…”
Line 114 – Including the drainage area for the entire contiguous watershed as well as for each of the 8 tributary streams would add valuable information to the manuscript regarding the sampling area.
Line 122 – Insert a space after the comma and before “Google.”
Line 123 – Please eliminate the extraneous spaces before “resulting.”
Line 133 - Please insert a space before “BHX.” Also, no explanation is provided regarding the absence of a site 1 in ZHX. Please add an explanation to address this.
Line 136 – “Surber” is missing the “r.” Please amend this.
Line 137 – It appears there may be an extra space before the comma.
Line 139 – Please expand the statement regarding how the substrate was disturbed. Was this done by hand or by brush or by some other method?
Line 144 – Was “most precise taxonomic unit possible” based on established taxonomy or authors’ ability to identify? Also, include a statement and relevant references to identify the resources used to identify macroinvertebrates. Also provide your data that show low taxonomic level of identification (i.e., family, genus, and/or species).
Line 160-162 – The sentence “For data sets…Kruskal-Wallis test” is redundant as this information is already conveyed in the sentence that immediately precedes it.
Line 214 – Please explain what “four-segmented mayflies” are. Are you intending to refer to tarsal segments, body segments, or other? Addressing this concern also hearkens back to the earlier statement regarding the importance of identifying your taxonomic resources.
Line 220 – The data reported in Fig. 3 only show order-level identification, in contrast to the claim in Line 144 about “most precise taxonomic unit possible.”
Line 229 – Pielou should be capitalized.
Line 254 – Add “was” after “which.”
Lines 266-67 – You mention “numerous studies” here but only cite 2. Please either amend the wording or include more relevant references to validate the claim.
Line 338 – Please clarify what “clean species” are. This presumably refers to species that are sensitive to pollution but “clean species” can be construed in a variety of ways (i.e., grooming habits of the insects, lack of sediment adhering to their exoskeleton, etc.).
Line 343 – Here again you have referenced “other studies” but only cited a single study. Please amend the wording or the included citations. Also does “the research” also refer back to this single citation? If so, there is another reason to include more relevant citations.
Line 345 – The phrase “large benthic invertebrates” is not clearly defined and as such, is entirely open to interpretation by the reader. Please provide some quantitative and/or descriptive parameters for what you regarded as large during the study.
Line 355 – Here again only a single study is cited to validate a claim about “existing studies.”
Line 356-58 – I question the legitimacy of the statement here, as it argues for a strictly linear relationship (i.e., increasing disturbance always and only leads to declining diversity). The seminal disturbance theories including Huston’s Dynamic Equilibrium Hypothesis and Connell’s Intermediate Disturbance Hypothesis argue that some levels of disturbance increase diversity by limiting dominant species in some conditions. Follow-up studies in the decades since have also continued this debate. If the paper you are citing here contradicts that claim, it is still important to acknowledge that there are differing perspectives on the ultimate impact of disturbance on diversity, such as what is reported in the Roxburgh et al. study you cited earlier.
Line 365 – Use of the phrase “the endemic taxon” implies that there is only one and that there can only be one. I question the legitimacy of the latter claim.
Line 481 – The journal The Science of the Total Environment is not fully capitalized here (and elsewhere throughout the list of references). However, there do not appear to be any formatting guidelines dictating that it should not be capitalized like all of the other journal titles. Please amend in all instances.
Line 467 – The journal title is not included for this reference. Please add it.
Line 484 – This reference is in all caps, presumably from a citation importation site. Please amend to follow the journal’s guidelines for references.
Line 487 – Please include the journal title for this reference.
Line 504 – The “Journal of Animal Ecology” is not fully capitalized here. Please amend.
Line 536 – The reference here is also in all caps. Please amend.
Comments on the Quality of English Language
The manuscript is reasonably well-written. Only minor revisions in wording and phrasing are needed.
Author Response
|
Responses to the first reviewer's comments |
|
Point-by-point response to Comments and Suggestions for Authors |
|
Comments 1: [Line 26 – A space is needed after the colon and “This” does not need to be in bold.] |
|
Response 1: Thank you for pointing this out. I have made changes to this. |
|
Comments 2: [Lines 60 – 62 – This is presently an incomplete sentence. Also, the statement incorrectly lumps flighted adult insects in with macro-invertebrates. Please note that following metamorphosis to the flighted adult stage, many formerly aquatic macroinvertebrates are no longer in a life stage that can be termed “benthic.” Please amend the wording to resolve both concerns.] |
|
Response 2: Agree. [I changed to “macro-invertebrates are aquatic organisms that spend all or part of their life history on the bottom surface or substrate of a body of freshwater (both flowing and standing) water, and whose individuals are unable to pass through a 425 μm (40 mesh) mesh screen. The highest diversity is found in aquatic insects, most of whose species are able to cross both land and water interfaces and exhibit flight characteristics”.] |
|
Comments 3: [Line 69 – Because of the usage here, “species” needs an apostrophe after it.] |
|
Response 3: Agree. I have made changes to this. |
|
Comments 4: [Line 80 – The phrase “in terms of biology” is superfluous and can be eliminated.] |
|
Response 4: Agree. I have made changes to this. |
|
Comments 5: [Line 85 – The phrase “various ways for the dispersal” is awkward. I recommend changing this to “various methods of dispersal known among…”] |
|
Response 5: Agree. I have made changes to this. |
|
Comments 6: [Lines 90 – 91 – As this sentence is currently worded it is not clear whether “their own” refers to “aquatic insects” or “topographical factors.” It is presumably the former, but the present phrasing leaves this ambiguous. I recommend changing this to say “…and the insects’ flight capabilities…”] |
|
Response 6: Agree. I have made changes to this.[Macro-invertebrates are aquatic organisms that spend all or part of their life history on the bottom surface or substrate of a body of freshwater (both flowing and standing) water, and whose individuals are unable to pass through a 425 μm (40 mesh) mesh screen. The highest diversity is found in aquatic insects, most of whose species are able to cross both land and water interfaces and exhibit flight characteristics.] |
|
Comments 7: [Line 114 – Including the drainage area for the entire contiguous watershed as well as for each of the 8 tributary streams would add valuable information to the manuscript regarding the sampling area.] |
|
Response 7: Agree. I have added the watershed area list. [Area of three land use types in each basin (Table 1)] |
|
Comments 8: [Line 122 – Insert a space after the comma and before “Google.”.] |
|
Response8: Agree. I have made changes to this. |
|
Comments 9: [Line 123 – Please eliminate the extraneous spaces before “resulting.”] |
|
Response 9: Agree. I have made changes to this. |
|
Comments 10: [Line 133 – Please insert a space before “BHX.” Also, no explanation is provided regarding the absence of a site 1 in ZHX. Please add an explanation to address this.] |
|
Response 10: Agree. I have made changes to this. [Because the upper neutralization stream (ZHX-1) was too steep to reach by humans.] |
|
Comments 11: [Line 136 – “Surber” is missing the “r.” Please amend this.] |
|
Response 11: Agree. I have made changes to this. |
|
Comments 12: [Line 137 – It appears there may be an extra space before the comma.] |
|
Response 12: Agree. I have made changes to this. |
|
Comments 13: [Line 139 – Please expand the statement regarding how the substrate was disturbed. Was this done by hand or by brush or by some other method?] |
|
Response 13: Agree. I have made changes to this. [The collection process consisted of hand sweeping macro-invertebrates on the stone surface into the net and then hand gently stirring the substrate for the macro-invertebrates to enter the net with the current.] |
|
Comments 14: [Line 144 – Was “most precise taxonomic unit possible” based on established taxonomy or authors’ ability to identify? Also, include a statement and relevant references to identify the resources used to identify macroinvertebrates. Also provide your data that show low taxonomic level of identification (i.e., family, genus, and/or species).] |
|
Response 14: [I have shown the latest taxon in the appendix. Macro-invertebrates were identified to the level of the genus or species.] |
|
Comments 15: [Line 160 – 162 – The sentence “For data sets…Kruskal-Wallis test” is redundant as this information is already conveyed in the sentence that immediately precedes it.] |
|
Response 15: [Has been deleted.] |
|
Comments 16: [Line 214 – Please explain what “four-segmented mayflies” are. Are you intending to refer to tarsal segments, body segments, or other? Addressing this concern also hearkens back to the earlier statement regarding the importance of identifying your taxonomic resources.] |
|
Response 16: [Change to the scientific name, Baetis sp1.] |
|
Comments 17: [Line 220 – The data reported in Fig. 3 only show order-level identification, in contrast to the claim in Line 144 about “most precise taxonomic unit possible.”.] |
|
Response 17: Agree. I have made changes to this. |
|
Comments 18: [Line 229 – Pielou should be capitalized.] |
|
Response 18: Agree. I have made changes to this. |
|
Comments 19: [Line 254 – Add “was” after “which.”] |
|
Response 19: Agree. I have made changes to this. |
|
Comments 20:[Lines 266 – 67 – You mention “numerous studies” here but only cite 2. Please either amend the wording or include more relevant references to validate the claim.] |
|
Response 20: Agree. I have made changes to this. |
|
Comments 21: [Line 338 – Please clarify what “clean species” are. This presumably refers to species that are sensitive to pollution but “clean species” can be construed in a variety of ways (i.e., grooming habits of the insects, lack of sediment adhering to their exoskeleton, etc.).] |
|
Response 21: [Change to a sensitive taxa.] |
|
Comments 22: [Line 343 – Here again you have referenced “other studies” but only cited a single study. Please amend the wording or the included citations. Also does “the research” also refer back to this single citation? If so, there is another reason to include more relevant citations.] |
|
Response 22: Agree. I have made changes to this. |
|
Comments 23: [Line 345 – The phrase “large benthic invertebrates” is not clearly defined and as such, is entirely open to interpretation by the reader. Please provide some quantitative and/or descriptive parameters for what you regarded as large during the study.] |
|
Response 23: [Added in the introduction. Macro-invertebrates are aquatic organisms that spend all or part of their life history on the bottom surface or substrate of a body of freshwater (both flowing and standing) water, and whose individuals are unable to pass through a 425 ㎛ (40 mesh) mesh screen.] |
|
Comments 24: [Line 355 – Here again only a single study is cited to validate a claim about “existing studies.”] |
|
Response 24: Agree. I have made changes to this. |
|
Comments 25: [Line 356-58 – I question the legitimacy of the statement here, as it argues for a strictly linear relationship (i.e., increasing disturbance always and only leads to declining diversity). The seminal disturbance theories including Huston’s Dynamic Equilibrium Hypothesis and Connell’s Intermediate Disturbance Hypothesis argue that some levels of disturbance increase diversity by limiting dominant species in some conditions. Follow-up studies in the decades since have also continued this debate. If the paper you are citing here contradicts that claim, it is still important to acknowledge that there are differing perspectives on the ultimate impact of disturbance on diversity, such as what is reported in the Roxburgh et al. study you cited earlier.] |
|
Response 25: [Our study only qualitatively addressed the human disturbance and did not conduct a quantitative analysis, which was not rigorously formulated in the previous manuscript. To this end, I will focus in the new manuscript on the diversity of macro-invertebrates in disturbed and undisturbed situations.] |
|
Comments 26: [Line 365 – Use of the phrase “the endemic taxon” implies that there is only one and that there can only be one. I question the legitimacy of the latter claim.] |
|
Response 26: Agree. I have made changes to this. |
|
Comments 27: [Line 481 – The journal The Science of the Total Environment is not fully capitalized here (and elsewhere throughout the list of references). However, there do not appear to be any formatting guidelines dictating that it should not be capitalized like all of the other journal titles. Please amend in all instances.] |
|
Response 27: Agree. I have made changes to this. [Science of The Total Environment] |
|
Comments 28: [Line 467 – The journal title is not included for this reference. Please add it.] |
|
Response 28: Agree. I have made changes to this. |
|
Comments 29: [Line 484 – This reference is in all caps, presumably from a citation importation site. Please amend to follow the journal’s guidelines for references.] |
|
Response 29: Agree. I have made changes to this. |
|
Comments 30: [Line 487 – Please include the journal title for this reference.] |
|
Response 30: Agree. I have made changes to this. |
|
Comments 31: [Line 504 – The “Journal of Animal Ecology” is not fully capitalized here. Please amend.] |
|
Response 31: Agree. I have made changes to this. [The Journal of Animal Ecology] |
|
Comments 32: [Line 536 – The reference here is also in all caps. Please amend.] |
|
Response 32: Agree. I have made changes to this. [Roxburgh, S.H.; Shea, K.; Wilson, J. The intermediate disturbance hypothesis: Patch dynamics and mechanisms of species coexistence. Ecology. 2004, 85, 359-371.] |

Reviewer 2 Report
Comments and Suggestions for Authors
1. In your Simple Summary (17-25) you mention : “In areas with low levels of anthropogenic activity, species distribution is limited by watershed segmentation. In areas with high anthropogenic disturbance, environmental factors showed homogeneity among streams, and species distribution was influenced by latitudinal stream connectivity with a more similar structure. “ but where latitudinal stream connectivity with a more similar structure is analysed in this paper.
2. In the Abstract (26-39) it is mentioned: a) “The results demonstrate macroinvertebrate community diversity positively correlated with forest land types (See Table 2 Margalef index with FL and CL and UL)”, [M1] b) “Key environmental factors influencing community structure included water temperature, pH, chlorophyll a, and oxidation-reduction potential (ORP).This phrase is not the same if we take into consideration PCA and RDA. C) In regions with low anthropogenic disturbance, species distribution was constrained by watershed topography.” Where in your analysis watershed topography is analysed? d[M2] ) “Conversely, high anthropogenic disturbance led to homogenized environmental conditions among adjacent streams (not always, see RDA), influencing species distribution through latitudinal stream connectivity (where does this is analysed in the paper?) and resulting in more uniform community structures.” E) “This study emphasizes the need for a comprehensive understanding of the interplay between land use changes, stream environments, and macroinvertebrate diversity to assess the ecological repercussions of human-induced disturbances in aquatic ecosystems (ecological repercussions of human-induced disturbances in aquatic ecosystems may be shown better by using biotic indices).
3. 96. Why parallel streams may describe better the dispersal of macroinvertebrates if they are not connected?
4. 100. The hypotheses are not clear in relation to the analysis and results that follow.
5. 131. The WHX:Wanhua Creek does not appear in the map of Fig.1 although it appears in the PCA and RDA. So where is it?
6. 2.3.2. You mention Margalef: How were organized the data as absolute numbers or as a density data matrix? (https://doi.org/10.1016/j.ecolind.2009.07.006)
7. 198-201. Only [M3] Fig. 2 reveals that the forest land (FL) and ORP for the upstream sites are the primary influencing factors contributing to environmental heterogeneity between upstream and all the other reaches ““ Flow velocity influences only site BSX-3 (Fig.2,5).
8. In Fig. 2 abbreviations must be described or in the legend they must refer to paragraph 2.2.2. (although in this paragraph salinity, forest Land, Cropland and Urban Land are not given as an abbreviation). Additionally, BE CAREFUL WHX-1,-2-3 DOES NOT APPEAR IN Fig. 1.
9. 3.2.1. In order to identify which taxa contributed most among the three reaches a Similarity Percentages Analysis (SIMPER) (Clarke et al., 2014) must be performed, with a cut-off of 90%. So will be shown similarities and dissimilarities as to Average Abundance – Percentage of Cumulative Contribution .
10. The colors of Fig 3 are not apparent for example the Plecoptera. A table would be better.
11. Paragraph 3.2.2. is not clear. You mention:
“There was no significant difference in the diversity index of macroinvertebrates in the upper, middle and lower reaches (P > 0.05). However, the regional diversity index [M4] and the richness one were higher in the upper reaches [M5] than those in the middle and lower reaches. They were also [M6] more diverse than those in the mid and downstream [M7] reaches. Notably, both Species Richness and the Margalef index displayed positive correlations with woodland land use.The Shannon-Wiener (was correlated in the case of urban Land, Table 2) and Pielou indices [M8] were not correlated with the three reaches ( Table 2). The above shows that the upper reaches had [M9] [M10] a higher macroinvertebrate abundance and number of species composition. [M11] [M12]
12. 3.3. Describe the main steps of RDA (accepting that the length of gradient of DCA in the first axis < 3) PREFERABLY in the paragraph 2.3.1. since:
Nothing is mentioned as to the selection of environmental factors in RDA. They MUST satisfy the condition n-1 (where n is the number of sampled stations), present a P-Value less than 0.05 and their inflation factor must be smaller than 20.
For RDA analysis it must also be written the % relation of species with the environmental parameters and the P of axes (the 1st and all the canonical axes together) for the depiction of the data. If the P of all together is >0.05 then the depiction is not correct. The P values and the eigenvalues would be better to appear in a small table at the up right hand side (or elsewhere) of Figures 2.
13. The analysis of Fig. 2 and 5 is not well understood and consequently no sufficient and appropriate explanations are given in connection always to Table 1.
The authors say after the PCA analysis:
“However, adjacent sample sites exhibited a high degree of environmental similarity. The areas of the midstream and downstream polygons intersected, indicating greater environmental similarity between these areas. The combined analysis of Table 1 and Fig. 2+5 reveals that the forest land (FL) and ORP for the upstream sites are [M13] the primary influencing factors contributing to environmental heterogeneity between upstream and all the other reaches “ Flow velocity influences only site BSX-3 (Fig.2 and 5) . Actually, the two figures coincide as to their results.
14. As to RDA it seems (It must also be given the correlation of the 1 and 2nd axis with the main environmental factors) in Fig.5 that PC1 is correlated to WT,Crop Land, and FL, and PC2 with TDS, Sal and ORP ,Width. But if we study Table 1 and the environmental factors of the three reaches we notice that the mean values of almost all the main parameters (as to RDA) are lower at the downstream reach except for ORP. Although no statistical differences were found in the environmental factors it is noticeable that DO IS LOWER IN THE UPSTREAM REACH!!!!. How this is explained?
15. Table 2. What exactly is it correlated between forest land and cropland and urban area?.
16. Paragraph 3.3. is not correct. Overlap does not discriminate the two reaches. Only the upstream reach is discriminated due to Forest Land. THERE IS NO DIFFERNCE AS TO PCA and RDA, BETWEEN FIG.2. AND 5. See also point 5.
In 3.3. it is mentioned: “This discrepancy signifies notable distinctions in macroinvertebrate community structures between the upstreamand the other two segments. Forest Land and Crop land discriminated the upstream from the mid and downstream macroinvertebrate [M14] communities. Within the upstream community, the variance was primarily driven by woodland land use types and ORP. Conversely, the variance within the midstream and downstream communities was predominantly influenced by [M15] TDS, Salinity, Crop Land,temperature and width (especially the sites JX2-3, WHX-2 and SYX-2). Remarkably, the macroinvertebrate communities in some contiguous sample sites of the middle and lower reaches demonstrated consistent patterns in the RDA plots, underscoring a heightened similarity in communities among closely situated sample sites.”
17. Figure 5. Abbreviations must be described or in the legend they must refer to paragraph 2.2.2. (although in this paragraph salinity, forest Land, Cropland and Urban Land are not given as an abbreviation). [BE CAREFUL WHX-1,-2-3 DOES NOT APPEAR IN Fig. 1]
18. In the discussion and Conclusions corrections/proposals are made in the paper:
265. as depicted in Fig.2 must be erased.
288. However, must be deleted
291. urban area, and agricultural land area must be erased
294. the first hypothesis stands but not the second!!!!!!!!!!!!!!!!!
307. WHY they exhibit distinct geographical barriers???? Where this is clarified in the paper?
319. What does it mean the watershed isolation???? what isolation is described in this paper???
328. why latitudinal connectivity of rivers???? Where this is clarified in the paper?
345. why large macroinvertebrates???? Do you mean macroinvertebrates?
345. the 3d hypothesis says something else "the interaction between environmental factors and channel connectivity facilitates homogenous dispersal of organisms downstream"
353. why large invertebrates????Do you mean macroinvertebrates?
366-367 Limnodrylus and Physa must be written in italic.
373. why watershed partitioning???? Where this is clarified in the paper?
375. where a branching distribution of river channels occur in this study???
378-380. The last phrase is not right. It depends on the analysis, PCA or RDA.
381-386. n The last paragraph of Conclusions is not concluded from this study. It is better to be erased. And in 384. substrate is not examined in this paper.
19. In Appendix please add a column stating which of these taxa are sensitive to pollution/disturbance.

Author Response
For research article
Manuscript ID: insects-2831758
Title: The Structure of Macroinvertebrate Communities is Significantly Shaped
by the Partitioning of Watersheds and the Connectivity of Water Systems in
Regions Influenced by Human Activities
Authors: Rui Li, Xianfu Li, RongLong YANG, Muhammad Farooq, Zhen TIAN, Yaning
XU, Nan SHAO, Shuoran LIU *, Wen XIAO *
|
Response to the second reviewer's comments |
|
Point-by-point response to Comments and Suggestions for Authors |
|
Comments 1: [In your Simple Summary (17-25) you mention: “In areas with low levels of anthropogenic activity, species distribution is limited by watershed segmentation. In areas with high anthropogenic disturbance, environmental factors showed homogeneity among streams, and species distribution was influenced by latitudinal stream connectivity with a more similar structure. “ but where latitudinal stream connectivity with a more similar structure is analysed in this paper.] |
|
Comments 2: [In the Abstract (26-39) it is mentioned: a) “The results demonstrate macroinvertebrate community diversity positively correlated with forest land types (See Table 2 Margalef index with FL and CL and UL)”, [M1] b) “Key environmental factors influencing community structure included water temperature, pH, chlorophyll a, and oxidation-reduction potential (ORP).This phrase is not the same if we take into consideration PCA and RDA. C) In regions with low anthropogenic disturbance, species distribution was constrained by watershed topography.” Where in your analysis watershed topography is analysed? d[M2] ) “Conversely, high anthropogenic disturbance led to homogenized environmental conditions among adjacent streams (not always, see RDA), influencing species distribution through latitudinal stream connectivity (where does this is analysed in the paper?) and resulting in more uniform community structures.” E) “This study emphasizes the need for a comprehensive understanding of the interplay between land use changes, stream environments, and macroinvertebrate diversity to assess the ecological repercussions of human-induced disturbances in aquatic ecosystems (ecological repercussions of human-induced disturbances in aquatic ecosystems may be shown better by using biotic indices).] |
|
Response 2: [This study explores the impact of anthropogenic land use changes on macro-invertebrates community structure in the streams of Cangshan Mountain. Through field collections of macro-invertebrates, measurement of water environments, and delineation of riparian zone land use in eight streams, we analysed the relationship between land use types, stream water environments, and macro-invertebrates diversity. The results demonstrate macro-invertebrates community diversity positively correlated with forest land types but negatively correlated with croplands and urban areas. Urban Land use type and water temperature is the main environmental factor causing the differences in macro-invertebrates communities upstream and middle and downstream. The disturbed streams had lower aquatic biodiversity than those in their natural state, showing a decrease in aquatic insect-susceptible taxa and a more similar community structure. In the natural woodland area,, species distributions may be constrained by watershed segmentation, present more complex community characteristics.] |
|
Comments 3: [96. Why parallel streams may describe better the dispersal of macroinvertebrates if they are not connected?] |
|
Response 3: [Our study assumes that in the Latitudinal direction upstream, geographic barriers with a ridge may well explain the dispersal of flight-competent species with distance.] |
|
Comments 4: [100. The hypotheses are not clear in relation to the analysis and results that follow.] |
|
Response 4: [Have made changes to the assumptions. (1) In the headwaters, the geographic isolation that exists between streams can impede the dispersal of environmental factors and aquatic communities, thus allowing them to exhibit heterogeneity. (2) In the middle and lower reaches, anthropogenic disturbances may alter the connectivity of streams, allowing for more even dispersal between biotic and abiotic ones, thus allowing both the stream water environment and aquatic communities to exhibit homogeneity.] |
|
Comments 5: [131. The WHX: Wanhua Creek does not appear in the map of Fig.1 although it appears in the PCA and RDA. So where is it?] |
|
Response 5: [At the position above the sample point of Figure 1.] |
|
Comments 6: [2.3.2. You mention Margalef: How were organized the data as absolute numbers or as a density data matrix? (https://doi.org/10.1016/j.ecolind.2009.07.006)] |
|
Comments 7: [198-201. Only [M3] Fig. 2 reveals that the forest land (FL) and ORP for the upstream sites are the primary influencing factors contributing to environmental heterogeneity between upstream and all the other reaches “ Flow velocity influences only site BSX-3 (Fig.2,5).] |
|
Response 7: [Modifications have been made. FL and ORP at upstream sites are the main influencing factors affecting environmental heterogeneity in upstream and other reaches. The flow rate only affects the site BSX-3 (Fig.3).] |
|
Comments 8: [In Fig. 2 abbreviations must be described or in the legend they must refer to paragraph 2.2.2. (although in this paragraph salinity, forest Land, Cropland and Urban Land are not given as an abbreviation). Additionally, BE CAREFUL WHX-1,-2-3 DOES NOT APPEAR IN Fig. 1.] |
|
Response 8: [Description has been added. WT: water temperature, Cond: conductivity, DO: dissolved oxygen, ORP: oxidation-reduction potential, TDS: total dissolved solids, Sal: salinity, Width: stream width, FV: flow velocity, Chla: chlorophyll-a, FL: Forest Land, UL: Urban Land, CL: Crop Land.] |
|
Comments 9: [3.2.1. In order to identify which taxa contributed most among the three reaches a Similarity Percentages Analysis (SIMPER) (Clarke et al., 2014) must be performed, with a cut-off of 90%. So will be shown similarities and dissimilarities as to Average Abundance – Percentage of Cumulative Contribution.] |
|
Response 9: [Table 4 has been added to illustrate this issue. Through SIMPER analysis , several species with relatively different contributions to the upstream, midstream and downstream community aggregation were identified.] |
|
Comments 10: [The colors of Fig 3 are not apparent for example the Plecoptera. A table would be better.] |
|
Response 10: [Has been changed to a more detailed taxa. Place in the annex section at the end of the manuscript.] |
|
Comments 11: Paragraph 3.2.2. is not clear. You mention: “There was no significant difference in the diversity index of macroinvertebrates in the upper, middle and lower reaches (P > 0.05). However, the regional diversity index [M4] and the richness one were higher in the upper reaches [M5] than those in the middle and lower reaches. They were also [M6] more diverse than those in the mid and downstream [M7] reaches. Notably, both Species Richness and the Margalef index displayed positive correlations with woodland land use. The Shannon-Wiener (was correlated in the case of urban Land, Table 2) and Pielou indices [M8] were not correlated with the three reaches ( Table 2). The above shows that the upper reaches had [M9] [M10] a higher macroinvertebrate abundance and number of species composition. [M11] [M12]] |
|
Response 11: [Description has been added. Differential analysis of the diversity index of the upper, middle and lower macro-invertebrates revealed no significant differences (P > 0.05). However, the upstream diversity index was larger than the middle and downstream, Richness indexes showed the trend of upstream > middle > downstream, Shannon-Wiener index and Pielou showed the upstream > downstream > middle trend (Fig. 4 ). This trend suggests that the upper reaches are species-rich hotspots, while the middle and lower reaches have relatively low macro-invertebrates abundance and simple species composition. Notably, Species Richness displayed positive correlations with woodland land use, while the Shannon-Wiener index showcased a negative correlation with urban land use (Table. 5 ).] |
|
Comments 12: [3.3. Describe the main steps of RDA (accepting that the length of gradient of DCA in the first axis < 3) PREFERABLY in the paragraph 2.3.1. since: Nothing is mentioned as to the selection of environmental factors in RDA. They MUST satisfy the condition n-1 (where n is the number of sampled stations), present a P-Value less than 0.05 and their inflation factor must be smaller than 20. For RDA analysis it must also be written the % relation of species with the environmental parameters and the P of axes (the 1st and all the canonical axes together) for the depiction of the data. If the P of all together is >0.05 then the depiction is not correct. The P values and the eigenvalues would be better to appear in a small table at the up right hand side (or elsewhere) of Figures 2.] |
|
Response 12: [Before selecting the appropriate correlation analysis means, it is necessary to conduct the Detrended Correspondence Analysis (DCA) of the biological community data, and determine whether the community structure is a unimodal model or a linear model. The DCA analysis showed that the FLG value (Final Length of Gradient, FLG) was 2.017. Screening of environmental factors: Collinearity analysis was performed for all environmental factors. The largest variable is deleted in turn, namely the environmental factor of collinearity, until all the variables are less than 10. We then detect the lowest AIC(Akaike information criterion) value with the step model, and in this step the model automatically screens out the optimal environmental factor. The environmental factors we included in the RDA analysis were: water temperature (WT), conductivity (Cond), dissolved oxygen (DO), oxidation-reduction potential (ORP), flow velocity (FV) and urban land (UL) (R 4.3.1).] |
|
Comments 13: [The analysis of Fig. 2 and 5 is not well understood and consequently no sufficient and appropriate explanations are given in connection always to Table 1. The authors say after the PCA analysis: “However, adjacent sample sites exhibited a high degree of environmental similarity. The areas of the midstream and downstream polygons intersected, indicating greater environmental similarity between these areas. The combined analysis of Table 1 and Fig. 2+5 reveals that the forest land (FL) and ORP for the upstream sites are [M13] the primary influencing factors contributing to environmental heterogeneity between upstream and all the other reaches “ Flow velocity influences only site BSX-3 (Fig.2 and 5) . Actually, the two figures coincide as to their results.] |
|
Response 13: [Modifications have been made accordingly in the article. FL and ORP at upstream sites are the main influencing factors affecting environmental heterogeneity in upstream and other reaches. The flow rate only affects the site BSX-3 (Fig.3).] |
|
Comments 14: [As to RDA it seems (It must also be given the correlation of the 1 and 2nd axis with the main environmental factors) in Fig.5 that PC1 is correlated to WT, Crop Land, and FL, and PC2 with TDS, Sal and ORP ,Width. But if we study Table 1 and the environmental factors of the three reaches we notice that the mean values of almost all the main parameters (as to RDA) are lower at the downstream reach except for ORP. Although no statistical differences were found in the environmental factors it is noticeable that DO IS LOWER IN THE UPSTREAM REACH!!!!. How this is explained?] |
|
Response 14: [Because the water flow of the upper, middle and downstream is different, the downstream flow is large, the flow rate is fast, the water and air exchange rate is large, so the dissolved oxygen is higher than the upstream.] |
|
Comments 15: [Table 2. What exactly is it correlated between forest land and cropland and urban area?] |
|
Response 15: [The areas above the upstream sample points are all forest areas, and arable land and construction land is not allowed according to the local government policy. The middle and downstream sample sites are in the area where cultivated land and construction land are alternately nested.] |
|
Comments 16: [Paragraph 3.3. is not correct. Overlap does not discriminate the two reaches. Only the upstream reach is discriminated due to Forest Land. THERE IS NO DIFFERNCE AS TO PCA and RDA, BETWEEN FIG.2. AND 5. See also point 5. In 3.3. it is mentioned: “This discrepancy signifies notable distinctions in macroinvertebrate community structures between the upstream and the other two segments. Forest Land and Crop land discriminated the upstream from the mid and downstream macroinvertebrate [M14] communities. Within the upstream community, the variance was primarily driven by woodland land use types and ORP. Conversely, the variance within the midstream and downstream communities was predominantly influenced by [M15] TDS, Salinity, Crop Land, temperature and width (especially the sites JX2-3, WHX-2 and SYX-2). Remarkably, the macroinvertebrate communities in some contiguous sample sites of the middle and lower reaches demonstrated consistent patterns in the RDA plots, underscoring a heightened similarity in communities among closely situated sample sites.”] |
|
Response 16: [The macro-invertebrates communities of the upstream, midstream, and downstream sections exhibited partial overlap on the RDA ordination map (Fig. 5). Urban Land use type and water temperature is the main environmental factor causing the differences in macro-invertebrates communities upstream and middle and downstream. In upstream, macro-invertebrates community differences are mainly driven by DO, and the differences in middle and lower communities are mainly affected by ORP, Cond and water flow velocity.] |
|
Comments 17: [Figure 5. Abbreviations must be described or in the legend they must refer to paragraph 2.2.2. (although in this paragraph salinity, forest Land, Cropland and Urban Land are not given as an abbreviation). [BE CAREFUL WHX-1,-2-3 DOES NOT APPEAR IN Fig. 1] |
|
Response 17: [Modifications have been made accordingly in the article. Eigenvalues values and the P-values of the axes. The total P-value of the RDA model was 0.028, WT: water temperature, Cond: conductivity, DO: dissolved oxygen, ORP: oxidation-reduction potential, FV: flow velocity, UL: Urban Land.] |
|
Comments 18: [In the discussion and Conclusions corrections/proposals are made in the paper: 265. as depicted in Fig.2 must be erased. 288. However, must be deleted 291. urban area, and agricultural land area must be erased 294. the first hypothesis stands but not the second!!!!!!!!!!!!!!!!! 307. WHY they exhibit distinct geographical barriers???? Where this is clarified in the paper? 319. What does it mean the watershed isolation???? what isolation is described in this paper??? 328. why latitudinal connectivity of rivers???? Where this is clarified in the paper? 345. why large macroinvertebrates???? Do you mean macroinvertebrates? 345. the 3d hypothesis says something else "the interaction between environmental factors and channel connectivity facilitates homogenous dispersal of organisms downstream" 353. why large invertebrates????Do you mean macroinvertebrates? 366-367 Limnodrylus and Physa must be written in italic. 373. why watershed partitioning???? Where this is clarified in the paper? 375. where a branching distribution of river channels occur in this study??? 378-380. The last phrase is not right. It depends on the analysis, PCA or RDA. 381-386. n The last paragraph of Conclusions is not concluded from this study. It is better to be erased. And in 384. substrate is not examined in this paper.] |
|
Response 18: [This has been revised in the article.] |
|
Comments 19: [In Appendix please add a column stating which of these taxa are sensitive to pollution/disturbance.] |
|
Response 19: Thank you for pointing this out. [I have added an appendix at the end of the article.] |

Reviewer 3 Report
Comments and Suggestions for Authors
Dear Authors,
Your paper is a valuable contribution to bioassessment of river condition using benthic macroinvertebrates. It is well written and informative. I have made recommendations and edits within the attached text. I am recommending that the paper be published, but with one major change. Your research does not support a study of connectivity. Rather the research is a simple bioassessment along different use regimes. In this regard, the paper is important and should be printed.
I recommend that you change the title from "The Structure of Macroinvertebrate Communities is Significantly Shaped by the Partitioning of Watersheds and the Connectivity of Water Systems in Regions Influenced by Human Activities" to "Bioassessment of Macroinvertebrate Communities Influenced by Gradients of Human Activities".
An example of a paper that explicitly studies the influence of connectivity on biodiversity may be found here, Maasri A, Gelhaus J (2012) Stream invertebrate communities of Mongolia: current structure and expected changes due to climate change. Aquatic Biosystems 8(1):18. doi:10.1186/2046-9063-8-18.
Also, your paper may be improved by including a tolerance index such as a Hilsenhoff Biotic Index to examine community changes related to tolerance in addition to impacts on diversity.
cheers

Author Response
|
Response to the 3rd reviewer's comments |
|
Point-by-point response to Comments and Suggestions for Authors |
|
Comments 1: [Your paper is a valuable contribution to bioassessment of river condition using benthic macroinvertebrates. It is well written and informative. I have made recommendations and edits within the attached text. I am recommending that the paper be published, but with one major change. Your research does not support a study of connectivity. Rather the research is a simple bioassessment along different use regimes. In this regard, the paper is important and should be printed.] |
|
Response 1: Thank you for pointing this out. [We focus our paper on the relationship between land-use patterns and macro-invertebrates communities, without emphasizing the issue of connectivity. Major changes have been made to the original manuscript.] |
|
Comments 2: [I recommend that you change the title from "The Structure of Macroinvertebrate Communities is Significantly Shaped by the Partitioning of Watersheds and the Connectivity of Water Systems in Regions Influenced by Human Activities" to "Bioassessment of Macroinvertebrate Communities Influenced by Gradients of Human Activities".] |
|
Response 2: Agree. We've changed the title accordingly to emphasize this point. [In line 2 of the updated manuscript] |
|
Comments 3: [An example of a paper that explicitly studies the influence of connectivity on biodiversity may be found here, Maasri A, Gelhaus J (2012) Stream invertebrate communities of Mongolia: current structure and expected changes due to climate change. Aquatic Biosystems 8(1):18. doi:10.1186/2046-9063-8-18.] |
|
Response 3: Thank you for pointing this out. We agree with this comment. To this and in the manuscript. |
|
Comments 4: [Also, your paper may be improved by including a tolerance index such as a Hilsenhoff Biotic Index to examine community changes related to tolerance in addition to impacts on diversity.] |
|
Response 4: Agree. We have added Table 3 to the manuscript to emphasize this change. [The mean FBI index is less than the middle and downstream (Table 3).] |
|
Comments 4: [Perhaps change to "adults that emerge into the terrestrial environment, . . ."] |
|
Response 4: [Change to:The highest diversity is found in aquatic insects, most of whose species are able to cross both land and water interfaces and exhibit flight characteristics.] |
|
Comments 5: [Change to "These patterns depend on the ecological niches . . ."; Change to "patterns related to . . ."] |
|
Response 5: [Change to:Depending on the strength of the adults' migratory ability and the larvae's drifting habits downstream, their community structure may show patterns related to different habitat conditions. These patterns depend on the ecological niches ecological niches of the biological groups themselves.] |
|
Comments 6: [patterns] |
|
Response 6: [In mountainous systems, these patterns are also related to the elevation gradient.] |
|
Comments 7: [Suggested change, "Alterered connectivity, in turn, may impact macroinvertebrate pathways for dispersal and the . . . "] |
|
Response 7: [Alterered connectivity, in turn, may impact macro-invertebrates pathways for dispersal" and the distribution of communities might change accordingly.] |
|
Comments 8: [Space] |
|
Response 8: [I have removed the whitespace.] |
|
Comments 9: [This map works well, but could you add in a larger map of China with the study area shown?] |
|
Response 9: [I added Figure 1] |
|
Comments 10: [Remove "Legend". Move tihs part up to the rest of the Figure legend above.] |
|
Response 10: [Changes have been completed in the manuscript.] |
|
Comments 11: [Please include net mesh aperature size unit.] |
|
Response 11: [Macro-invertebrates were collected using a Surber net with a pore size of 0.5 mm, featuring a sampling area of 0.09 m², positioned at the riverbed bottom aligned with the water flow direction.(Sampling replications is 5 times) ] |
|
Comments 12: [Move to end of sentence.] |
|
Response 12: [Upstream and midstream environmental data were analyzed for differences in SPSS software (version 25.0).] |
|
Comments 13: [Move to end of sentance.] |
|
Response 13: [Changes have been completed in the manuscript.] |
|
Comments 14: [Remove] |
|
Response 14: [Changes have been completed in the manuscript.] |
|
Comments 15: [I am unfamiliar with four-segmented mayflies and hooked shrimp. Perhaps use the scientific names for these taxa. And I think the dominant Diptera are chironomids (non-biting midges) within your study according to the appendix? Please use the scientific name.] |
|
Response 15: [The mean FBI index is less than the middle and downstream (Table 3). The table of relative abundance of species composition of macro-invertebrates in the upstream, midstream and downstream shows that Plecoptera occurs only in the upstream and that Kamimuria,Cryptoterla sp1. and Amphinemura sp1. has a low fouling tolerance value. The species composition of the upstream is generally that of species with low fouling tolerance values (Appendix).] |
|
Comments 16: [Change to "Due to . . . "] |
|
Response 16: [Due to the high forest cover in the upstream, although micro-scale environmental heterogeneity is lower than in the middle and lower reaches, the complexity of community structure remains significantly higher than in the middle and lower reaches.] |
|
Comments 17: [Perhaps this is true, but the analysis does not directly test connectivity. ] |
|
Response 17: [Elevation topographic maps of upstream sample points have been added to the updated manuscript, demonstrating that there is a geographic barrier between the upstream sample points.] |
|
Comments 18: [Remove.] |
|
Response 18: [Changes have been completed in the manuscript.] |
|
Comments 19: [macroinvertebrates] |
|
Response 19: [Changes have been completed in the manuscript.] |
|
Comments 20: [This is true, so I suggest including a tolerance metric in your study.] |
|
Response 20: [Macro-invertebrates tolerance values have been added to the updated manuscript and calculated for FBI.] |
|
Comments 21: [Collembola] |
|
Response 21: [In the updated manuscript, a more detailed level of categorization was added] |

Round 2
Reviewer 1 Report
Comments and Suggestions for Authors
General Comments for Authors
The edited version of the manuscript is improved. Specifically, clarification of the sampling methodology, the definition of macroinvertebrate, and the inclusion of the taxonomic information in the Appendix are helpful additions that strengthen the manuscript. Some minor editing is needed throughout the manuscript and specific recommendations are detailed below, including some issues that need to be addressed repetitively throughout the manuscript. One major component that still needs to be addressed is the inclusion of citations and references for the taxonomic work in the study. Without such references, the validity of taxonomic identification is unknown.
Specific Comments for Authors
Line 2 – You use “macroinvertebrates” in the plural form here, however it does not need to be. In this context (and in many similar situations elsewhere throughout the manuscript) the term is describing “communities,” so the singular form is the more appropriate use (i.e., “macroinvertebrate communities”). See, for example, https://doi.org/10.1046/j.1365-2427.2001.00758.x
Line 19 – The phrase “…aquatic insect-susceptible taxa…” is incorrect. As stated, this phrase seems to indicate that the taxa is liable to be attacked or infested by aquatic insects. I think you mean to indicate that there was a “decrease in disturbance-sensitive aquatic insect taxa…” Please amend here and in similar situations throughout the manuscript.
Line 20 – There is an extra comma after “area” that can be eliminated.
Line 21 – Between “segmentation” and “present” the comma should be replaced by “and.”
Line 28 – Amend this sentence to use the correct plural form (i.e., “Urban land use type and water temperature are the main environmental factors…”)
Line 32 – There is an extra comma after “area.”
Line 54 – Change “one” to “some,” since you are referring to ecosystems in the plural.
Line 60 – Here you hint at “another factor,” but you don’t explicitly state it. To clarify for the reader and to set the stage for the rest of the article, please state this other factor here.
Line 65- Here you introduce the acronym “FBI,” but you have neglected to define what this stands for like you did below in line 68 in describing the HBI. Please add the definition for the acronym.
Line 69 – There is an extraneous “index” at the end of this sentence that can be eliminated.
Line 73 – Your amended description in this section is improved, however I am uncertain about what you mean in claiming that “the highest diversity is found in aquatic insects.” Are you suggesting that aquatic insect species outnumber terrestrial insect species? I question the legitimacy of such a claim. Please either provide a reference to validate this claim or amend the wording if you are intending to convey something else.
Line 77 – The phrase “ecological niches” is unnecessarily included twice. Please eliminate one.
Line 81 – The addition of the apostrophe at the end of species is correct to show possession. However, the addition of the apostrophe at the beginning of the word is unnecessary and should be eliminated.
Line 93 – There is an unnecessary double apostrophe at the end of the word dispersal that can be eliminated.
Line 96 – In “…among of macroinvertebrates…” eliminate the word “of.”
Line 102 – Here an apostrophe after insects is still needed to show possession (i.e., “…the insects’ flight capabilities…”)
Line 115 – I am uncertain what “ones” refers to here. Is it referring to factors, communities, streams? I recommend using a more precise word to eliminate ambiguity.
Line 136 – You refer to amphibian populations here, so a reference is needed to validate this claim.
Line 154 – A space is needed after the colon.
Line 155 – The addition of Table 1 is informative and strengthens your manuscript.
Line 160 – The parenthetical insert appears to be a note-to-self and in a different font than the main text. I recommend making this an independent sentence (i.e., “Five replicates were collected at each site.”)
Line 160-62 – The addition of this description is helpful. Thank you for adding it.
Line 169 – Clarifying the taxonomic level is helpful, but you still haven’t included the resources you used to determine identification. This is a crucial component that must be included here and/or elsewhere in the manuscript to validate the identification process. Also, if only a few taxa were identified to the family level list them in a parenthetical insert.
Line 204 – The amended wording here appears to have been done quickly such that the sentence still reads like a note-to-self. Please amend the wording. Since you have already completed this step, it should be described in the past tense too, as you have done elsewhere throughout the manuscript.
Line 210-12 – Similarly, this sentence should be described in the past tense.
Line 246 – Use past tense.
Line 264 – The values in Table 4 are misaligned. Please amend this.
Line 324 – There is an extra period here that can be removed.
Line 340 – Eliminate the word “off” or reword to say “…blocked off horizontally…”
Line 342 – Here is an example where use of the word “macroinvertebrates” in the plural form is correct. (This is in contrast to the many instances in the manuscript where the word is used as a descriptor and should be in the singular form, as mentioned in the comment for Line 2.)
Line 361 – I recommend rephrasing to say “…reaches may lead to…” as this is a more cautious phrase and allows for unexpected consequences that may occur. Also, please define what “extended streams” are or use a different phrase.
Line 362 – I recommend rephrasing this sentence. The phrase “…as a group sensitive to environmental changes…” suggests that all macroinvertebrates are sensitive to pollution, but that is not accurate. Some, such as leeches, midges, black flies, etc. are tolerant of pollution, so macroinvertebrates cannot be lumped into a single ecologically responsive group.
Line 391-95 – Use past tense.
Line 396 – This sentence needs to be amended. I suggest “The Plecoptera found only…have a low…”
Line 401 – Same comment as in Line 19.
Line 402 – Add “and” before “present.”
Line 405 – Change “is” to “are.”
Line 406 – Change “factor” to the plural form, since you have reference two factors.
Line 513 – This reference is still in all capitals and, therefore, is inconsistent with the formatting used for the rest of the references. Please amend to conform to the journal’s guidelines for formatting references.
Line 599 – The addition of the appendix is very informative and strengthens the manuscript. The lack of any in-text citation and lack of any listed references declaring which taxonomic resources you used for identification, however, is still glaringly absent. Citation of taxonomic resources is a standard component of macroinvertebrate studies (see the manuscript referenced previously in the Line 2 comment wherein the authors have specifically cited and referenced at least four separate taxonomic resources). This situation must be remedied in your manuscript. The absence of this information calls into question the validity of some components of the study.
Comments on the Quality of English Language
The manuscript is reasonably well written. Some editing is needed to improve phrasing, and specific instances are pointed out in the specific comments for authors provided above.
Author Response
Comments 1: [Line 2- You use “macroinvertebrates” in the plural form here, however it does not need to be. In this context (and in many similar situations elsewhere throughout the manuscript) the term is describing “communities”, so the singular form is the more appropriate use (i.e., “macroinvertebrate communities”). See, for example, https://doi.org/10.1046/j.1365-2427.2001.00758.x]
Response 1: [Has made revisions in the manuscript.]
Comments 2: [Line 19 – The phrase “…aquatic insect-susceptible taxa…” is incorrect. As stated, this phrase seems to indicate that the taxa is liable to be attacked or infested by aquatic insects. I think you mean to indicate that there was a “decrease in disturbance-sensitive aquatic insect taxa…” Please amend here and in similar situations throughout the manuscript.]
Response 2: [The disturbed streams had lower aquatic biodiversity than those in their natural state, decrease in disturbance-sensitive aquatic insect taxa and a more similar community structure.]
Comments 3: [Line 20 – There is an extra comma after “area” that can be eliminated.]
Response 3: [Has made revisions in the manuscript.]
Comments 4: [Line 21 – Between “segmentation” and “present” the comma should be replaced by “and.”]
Response 4: [Has made revisions in the manuscript.]
Comments 5: [Line 28 – Amend this sentence to use the correct plural form (i.e., “Urban land use type and water temperature are the main environmental factors…”)]
Response 5: [Has made revisions in the manuscript.]
Comments 6: [Line 32 – There is an extra comma after “area.”]
Response 6: [Has made revisions in the manuscript.]
Comments 7: [Line 54 – Change “one” to “some,” since you are referring to ecosystems in the plural.]
Response 7: [Has made revisions in the manuscript.]
Comments 8: [Line 60 – Here you hint at “another factor,” but you don’t explicitly state it. To clarify for the reader and to set the stage for the rest of the article, please state this other factor here..]
Response 8: [However, the issue of the resulting dispersal distribution of species has been neglected and may be the root cause of community change in aquatic ecosystems disturbed by human activities]
Comments 9: [Line 65- Here you introduce the acronym “FBI,” but you have neglected to define what this stands for like you did below in line 68 in describing the HBI. Please add the definition for the acronym.]
Response 9: [Family biological index (FBI)]
Comments 10: [Line 69 – There is an extraneous “index” at the end of this sentence that can be eliminated]
Response 10: [Has made revisions in the manuscript.]
Comments 11: [Line 73 – Your amended description in this section is improved, however I am uncertain about what you mean in claiming that “the highest diversity is found in aquatic insects.” Are you suggesting that aquatic insect species outnumber terrestrial insect species? I question the legitimacy of such a claim. Please either provide a reference to validate this claim or amend the wording if you are intending to convey something else.]
Response 11: [Aquatic insects are an important part of the macro-invertebrate, most of whose species are able to cross both land and water interfaces and exhibit flight characteristics]
Comments 11: Line 77 – The phrase “ecological niches” is unnecessarily included twice. Please eliminate one.
Response 12: [Has made revisions in the manuscript.]
Comments 13: [Line 81 – The addition of the apostrophe at the end of species is correct to show possession. However, the addition of the apostrophe at the beginning of the word is unnecessary and should be eliminated.]
Response 13: [Has made revisions in the manuscript.]
Comments 14: [Line 93 – There is an unnecessary double apostrophe at the end of the word dispersal that can be eliminated.]
Response 14: [Has made revisions in the manuscript.]
Comments 15: [Line 96 – In “…among of macroinvertebrates…” eliminate the word “of.”]
Response 15: [Has made revisions in the manuscript.]
Comments 16: [Line 102 – Here an apostrophe after insects is still needed to show possession (i.e., “…the insects’ flight capabilities…”)]
Response 16: [Has made revisions in the manuscript.]
Comments 17: [Line 115 – I am uncertain what “ones” refers to here. Is it referring to factors, communities, streams? I recommend using a more precise word to eliminate ambiguity.]
Response 17: [ In the middle and lower reaches, anthropogenic disturbances may alter stream connectivity, resulting in more even dispersal between biotic and abiotic organisms, and thus homogeneity of the stream water environment and aquatic communities. ]
Comments 18: [Line 136 – You refer to amphibian populations here, so a reference is needed to validate this claim.]
Response 18: [For species with the potential to migrate over long distances, the presence of ridges resulted in low dispersal rates, which impeded inter-population gene flow, and may lead to high levels of differentiation in mountain macro-invertebrate populations.]
Comments 19: [Line 154 – A space is needed after the colon.]
Response 19: [Has made revisions in the manuscript.]
Comments 20: [Line 155 – The addition of Table 1 is informative and strengthens your manuscript.]
Response 20: [Thanks for your suggestions.]
Comments 21: [Line 160 – The parenthetical insert appears to be a note-to-self and in a different font than the main text. I recommend making this an independent sentence (i.e., “Five replicates were collected at each site.”)]
Response 21: [Five replicates were collected at each site.]
Comments 22: [Line 160-62 – The addition of this description is helpful. Thank you for adding it.]
Response 22: [Thanks for your suggestions.]
Comments 23: [Line 169 – Clarifying the taxonomic level is helpful, but you still haven’t included the resources you used to determine identification. This is a crucial component that must be included here and/or elsewhere in the manuscript to validate the identification process. Also, if only a few taxa were identified to the family level list them in a parenthetical insert.]
Response 23: [I have added three professional classified books as a reference.
51.Morse, J.C.; Yang, L.; Tian L. Aquatic Insects of China Useful for Monitoring Water Quality; Hohai University Press: Nanjing, China. 1994.
- Epler, J.H. Identification manual for the larval Chironomidae (Diptera) of North and South Carolina: a guide to the taxonomy of the midges of the southeastern United States, including Florida. North Carolina Department of Environmental and Natural Resources. St. John’s River Water Management District. Raleigh North Carolina. USA. 2001.
- Merritt R, W.; Cummins, K.W.; Berg, M.B. An Introduction to the Aquatic Insects of North America, fourth ed. Kendall and Hunt, Dubuque, Iowa, USA. 2008.]
Comments 24: [Line 204 – The amended wording here appears to have been done quickly such that the sentence still reads like a note-to-self. Please amend the wording. Since you have already completed this step, it should be described in the past tense too, as you have done elsewhere throughout the manuscript.]
Response 24: [Before selecting the appropriate means of correlation analysis, we performed a detrended correspondence analysis (DCA) on the biome data and determined whether the community structure was a unimodal or linear model.]
Comments 25: [Line 210-12 – Similarly, this sentence should be described in the past tense.]
Response 25: [Has made revisions in the manuscript.]
Comments 26: [Line 246 – Use past tense.]
Response 26: [Has made revisions in the manuscript.]
Comments 27: [Line 324 – There is an extra period here that can be removed.]
Response 27: [Has made revisions in the manuscript.]
Comments 28: [Line 340 – Eliminate the word “off” or reword to say “…blocked off horizontally…”]
Response 28: [The upstream site, inside the canyon, is blocked horizontally blocked off horizontally by a ridge, hindering the spread of flying species.]
Comments 29: [Line 342 – Here is an example where use of the word “macroinvertebrates” in the plural form is correct. (This is in contrast to the many instances in the manuscript where the word is used as a descriptor and should be in the singular form, as mentioned in the comment for Line 2.)]
Response 29: [Thanks for your suggestions]
Comments 30: [Line 361 – I recommend rephrasing to say “…reaches may lead to…” as this is a more cautious phrase and allows for unexpected consequences that may occur. Also, please define what “extended streams” are or use a different phrase.]
Response 30: [In the middle and lower reaches of the stream, human activities have influenced changes in water environmental factors. As a result, this has led to a decrease in environmentally sensitive taxa and an increase in pollution-tolerant taxa of macro-invertebrates]
Comments 31: [Line 362 – I recommend rephrasing this sentence. The phrase “…as a group sensitive to environmental changes…” suggests that all macroinvertebrates are sensitive to pollution, but that is not accurate. Some, such as leeches, midges, black flies, etc. are tolerant of pollution, so macroinvertebrates cannot be lumped into a single ecologically responsive group.]
Response 31: [In the middle and lower reaches of the stream, human activities have influenced changes in water environmental factors. As a result, this has led to a decrease in environmentally sensitive taxa and an increase in pollution-tolerant taxa of macro-invertebrates.]
Comments 32: [Line 391-95 – Use past tense.]
Response 32: [Has made revisions in the manuscript.]
Comments 33: [Line 396 – This sentence needs to be amended. I suggest “The Plecoptera found only…have a low…”]
Response 33: [The Plecoptera found only upstream, Kamimuria, Cryptoterla sp1. and Amphinemura sp1. has a low fouling tolerance value.]
Comments 34: [Line 401 – Same comment as in Line 19.]
Response 34: [Has made revisions in the manuscript.]
Comments 35: [Line 402 – Add “and” before “present.”]
Response 35: [In the natural woodland area, species distributions may be constrained by watershed segmentation, and present more complex community characteristics.]
Comments 36: [Line 405 – Change “is” to “are.”]
Response 36: [Urban Land use type and water temperature are the main environmental factors causing the differences in macro-invertebrate communities upstream and middle and downstream.]
Comments 37: [Line 406 – Change “factor” to the plural form, since you have reference two factors.]
Response 37: [Urban Land use type and water temperature are the main environmental factors causing the differences in macro-invertebrate communities upstream and middle and downstream.]
Comments 38: [Line 513 – This reference is still in all capitals and, therefore, is inconsistent with the formatting used for the rest of the references. Please amend to conform to the journal’s guidelines for formatting references.]
Response 38: [Pringle, C.M. Hydrologic connectivity and the management of biological reserves: a global perspective Ecological Applications : a Publication of the Ecological Society of America. 2001 Aug; 11(4): 981-998.]
Reviewer 2 Report
Comments and Suggestions for Authors
There are still changes to be done.

Author Response
|
Comments 1: [index was erased] |
|
Comments 2: [What role play the amphibian population to macroinvertebrates?] |
|
Response 2: [For species with the potential to migrate over long distances, the presence of ridges resulted in low dispersal rates, which impeded inter-population gene flow, and may lead to high levels of differentiation in mountain macro-invertebrate populations. ] |
|
Comments 3: [not between but among] |
|
Response 3: [Relationship among macro-invertebrate and basin-scale environmental factors] |
|
Comments 4: [erase ARE THAT] |
|
Response 4: [Possible reasons for these results are that in the middle and lower reaches, rivers pass through urban residential areas and agricultural buffer zones before entering lakes. ] |
|
Comments 5: [ This result is inconsistent with the first one proposed in this paper MUST BE ERASED BECAUSE IN THE 1ST HYPOTHESIS YOU ALSO MENTION AQUATIC COMMUNITIES WHICH ARE RICHER IN THE NUMBER OF SPECIES. SEE WHAT YOU MENTION "....impede the dispersal of environmental factors and aquatic communities, thus allowing them to exhibit heterogeneity......"] |
|
Response 5: [Have been removed in the manuscript.] |
|
Comments 6: [This suggests that the formation of upstream biological communities is influenced not only by environmental factors but also by watershed isolation. Watershed isolation restricts species dispersal, shaping the process of community establishment. ISOLATION PLAYS A DISADVANTAGEOUS ROLE TO THE DOWNSTREAM SITES BECUSE IT RESTRICKS DISPERSION.UPSTREAM biological communities ARE influenced BY environmental factors AND THE ABSENCE OF HUMAN ACTIVITIES.] |
|
Comments 7: [WILL MUST BE ERASED.] |
|
Response 7: [While species dispersal in space promotes species movement and colonization in new locations, environmental selection can still alter their community composition.] |
|
Comments 8: [IN EXTENDED STREAMS MUST BE ERASED.] |
|
Response 8: [In the middle and lower reaches of the stream, human activities have influenced changes in water environmental factors. As a result, this has led to a decrease in environmentally sensitive taxa and an increase in pollution-tolerant taxa of macro-invertebrates ] |
|
Comments 9: [HOWEVER MUST BE ERASED.] |
|
Response 9: [In the middle and lower reaches of the stream, human activities have influenced changes in water environmental factors. As a result, this has led to a decrease in environmentally sensitive taxa and an increase in pollution-tolerant taxa of macro-invertebrates |
|
Comments 10: [THE SECOND NOT THE THIRD] |
|
Response 10: [Have been revised in the manuscript.] |
|
Comments 11: [NOT LESS BUT LOWER] |
|
Response 11: [Have been revised in the manuscript.] |
|
Comments 12: [ERASE THE SECOND PHRASE "The mean FBI index is less than the middle and downstream. "] |
|
Response 12: [Have been revised in the manuscript.] |
|
Comments 13: [ERASE THE PHRASE "Species diversity displays a positive correlation with woodland land types and an 398 inversely proportional relationship with croplands and urban land." SINCE IT DOES NOT HELP AT ALL AND ADDS NOTHING.THE REST OF THE PARAGRAPH IS CLEAR AND ENOUGH.] |
|
Response 13: [Have been revised in the manuscript.] |
Round 3
Reviewer 1 Report
Comments and Suggestions for Authors
General Comments
The second round of revisions has led to further improvement of the manuscript. Wording has been improved and most of the recommended edits have been made. Some editing is still necessary, however, and specific suggestions are identified below. The concern about proper usage of the plural vs. singular form of “macroinvertebrate” was addressed, but the problem was over-corrected such that there are still many instances where the term is used incorrectly. These are also pointed out below. The addition of references for aquatic insect identification is an improvement, but it is still somewhat ambiguous whether these are resources that you used to identify the aquatic insects yourselves or whether these were merely added to bolster the list of references. Please clarify this. For example, “We identified macroinvertebrates to order and family level using (list appropriate references here) and then to genus and/or species level using (list appropriate references here).”
Specific Comments
Line 2 – You have included a hyphen within the word macroinvertebrate here and throughout the manuscript. In English, this word does not require a hyphen and typically a hyphen is not used. Unless the journal requires you to use a hyphen in the term, you can eliminate the hyphen here and throughout the manuscript.
Line 23-24 – Here is an example where the correct term should be the plural form (i.e., macroinvertebrates).
Line 26 – There appears to be an unnecessary comma after “demonstrate.” This can be eliminated.
Line 53 – “Family” does not need to be capitalized, since it is not the name of a person. If you choose to capitalize it since it is part of the title of the index, then you should capitalize the entire title (i.e., “Family Biological Index”). If you choose the latter, then you should do similarly in Line 56 for the Hilsenhoff Biological Index. Either is acceptable, but it is best to be consistent.
Line 58 – Macroinvertebrates (should be plural here)
Line 61 – Amend the phrasing here to say “… macroinvertebrate community…”
Line 70 - Macroinvertebrates (should be plural here)
Line 83 - Macroinvertebrates (should be plural here)
Line 95 - Macroinvertebrates (should be plural here)
Line 148 - Macroinvertebrates (should be plural here)
Line 149 – I recommend a slight change in the phrasing to say “…gently hand-stirring…”
Line 151 – There appear to be extraneous spaces before “Additionally” here that can be eliminated.
Line 153-54 – The addition of cited references here is helpful. However, it is still not clear to the reader whether you used these resources yourselves to determine the taxonomic identification of the insects. This must be clarified, as the study’s findings hinge primarily on accurate identification of the captured macroinvertebrate specimens. If you determined taxonomy yourselves, please clarify this (i.e., “We identified the macroinvertebrates to order and family level using (insert references here) and then to genus and species level using (insert references here).)” If you, instead, sent out your specimens to a professional laboratory for identification, please cite the laboratory that conducted the identification.
Line 183-84 - Macroinvertebrates (should be plural here)
Line 187 - Macroinvertebrates (should be plural here)
Line 234 - Macroinvertebrates (should be plural here)
Line 253 - Macroinvertebrates (should be plural here)
Line 266 - Macroinvertebrates (should be plural here)
Line 314-15 – The phrasing was changed here, but the phrase is now stated twice. Please eliminate the “…blocked horizontally…” from the sentence.
Line 317-18 - Macroinvertebrates (should be plural here)
Line 350 - Macroinvertebrates (should be plural here)
Line 358 - Macroinvertebrates (should be plural here) in both uses of the term
Line 361 - Macroinvertebrates (should be plural here)
Line 368 – Change “has” to “have,” since you are referring to multiple examples
Line 378 – Given the ongoing ambiguity regarding aquatic insect identification, it would be helpful to include in the “Author Contributions” section a clear statement about which authors conducted the aquatic insect identification.
Line 576 – A more descriptive title would be helpful here. Presumably, the list contains the aquatic macroinvertebrates you captured in your sampling. Therefore, “Macroinvertebrate taxa captured during sampling” is one suggestion for a more specific title. Also, if allowed by the journal, it would be helpful to more clearly separate the order name from the family and genus names. This could be done by bold print or underlining the order name. Also, you included some sub-families within the family column, so the heading should state “family or sub-family” to indicate this. There are also several mis-spellings in the taxonomic table including:
· For sp14 the correct family spelling is “Leptoceridae.”
· For sp26 the correct sub-family spelling is “Orthocladiinae.” Also, you listed the family name again under the genus column. Please correct this.
· For sp27 the sub-family is listed in the genus column too.
· For sp28 the sub-family is listed in the genus column too.
· For sp30 the stated order is undiscernible, but it looks like you intended to state “Deuterophlebiidae.” Also, there is only one genus “Deuterophlebia” for this family, so that spelling needs to be corrected too.
· For sp31 the family is listed in the genus column too.
· For sp47 you listed a genus in the family column. If the “Ephemera” genus here is correct, the family to which that genus belongs is “Ephemeridae.”
Line 577-79 – The phrasing “the number of species of this species…” is awkward. Do you mean the number of captured specimens in this species? If so, please amend the phrasing to reflect this.
Comments on the Quality of English Language
Use of the English language is decent and has improved since the first draft. However, some editing is still needed.
Author Response
Response to reviewer 1
Thank you very much for your suggestions, we have revised the manuscript according to your advices and made some further corrections. Please see the point-to-point response as follow:
Comments 1: [Line 2 – You have included a hyphen within the word macroinvertebrate here and throughout the manuscript. In English, this word does not require a hyphen and typically a hyphen is not used. Unless the journal requires you to use a hyphen in the term, you can eliminate the hyphen here and throughout the manuscript.]
Response 1: [Delete the hyphen in the full text.]
Comments 2: [Line 23-24 – Here is an example where the correct term should be the plural form (i.e., macroinvertebrates).]
Response 2: [Has made revisions in the manuscript.]
Comments 3: [Line 26 – There appears to be an unnecessary comma after “demonstrate.” This can be eliminated.]
Response 3: [Has made revisions in the manuscript.]
Comments 4: [Line 53 – “Family” does not need to be capitalized, since it is not the name of a person. If you choose to capitalize it since it is part of the title of the index, then you should capitalize the entire title (i.e., “Family Biological Index”). If you choose the latter, then you should do similarly in Line 56 for the Hilsenhoff Biological Index. Either is acceptable, but it is best to be consistent.]
Response 4: [We have revised the sentence as “The Family Biological Index (FBI) was proposed by the American scholar Hilsenhoff [17] in 1988. In order to reduce the difficulty of species identification and save time, and realize the rapid evaluation of river health, he proposed the FBI index on the basis of the Hilsenhoff Biological Index (HBI) established by him, which effectively promoted the wide application of Biological Index (BI)”]
Comments 5: [Line 58 – Macroinvertebrates (should be plural here)]
Response 5: [Has made revisions in the manuscript.]
Comments 6: [Line 61 – Amend the phrasing here to say “… macroinvertebrate community…”]
Response 6: [Has made revisions in the manuscript.]
Comments 7: [Line 70 - Macroinvertebrates (should be plural here)
Line 83 - Macroinvertebrates (should be plural here)
Line 95 - Macroinvertebrates (should be plural here)
Line 148 - Macroinvertebrates (should be plural here)]
Response 7: [Has made revisions in the manuscript.]
Comments 8: [Line 149 – I recommend a slight change in the phrasing to say “…gently hand-stirring…”]
Response 8: [The collection process consisted of hand sweeping macroinvertebrates on the stone surface into the net and then gently hand-stirring the substrate for the macroinvertebrates to enter the net with the current.]
Comments 9: [Line 151 – There appear to be extraneous spaces before “Additionally” here that can be eliminated.]
Response 9: [Has made revisions in the manuscript.]
Comments 10: [Line 153-54 – The addition of cited references here is helpful. However, it is still not clear to the reader whether you used these resources yourselves to determine the taxonomic identification of the insects. This must be clarified, as the study’s findings hinge primarily on accurate identification of the captured macroinvertebrate specimens. If you determined taxonomy yourselves, please clarify this (i.e., “We identified the macroinvertebrates to order and family level using (insert references here) and then to genus and species level using (insert references here).)” If you, instead, sent out your specimens to a professional laboratory for identification, please cite the laboratory that conducted the identification.]
Response 10: [We have revised the sentence as “The majority of macroinvertebrates were assigned to the genus or species level, with Chironomidae classified at the family or subfamily level, as per references [51-54]. Further-more, the macroinvertebrate specimens underwent verification by the authors responsible for the primary identifications.”]
Comments 11: [Line 183-84 - Macroinvertebrates (should be plural here)
Line 187 - Macroinvertebrates (should be plural here)
Line 234 - Macroinvertebrates (should be plural here)
Line 253 - Macroinvertebrates (should be plural here)
Line 266 - Macroinvertebrates (should be plural here)]
Response 11: [Has made revisions in the manuscript.]
Comments 12: [Line 314-15 – The phrasing was changed here, but the phrase is now stated twice. Please eliminate the “…blocked horizontally…” from the sentence.]
Response 12: [We have revised the sentence as “The upstream site, inside the canyon, is blocked off horizontally by a ridge, hindering the spread of flying species.”]
Comments 13: [Line 317-18 - Macroinvertebrates (should be plural here)
Line 350 - Macroinvertebrates (should be plural here)
Line 358 - Macroinvertebrates (should be plural here) in both uses of the term
Line 361 - Macroinvertebrates (should be plural here)]
Response 13: [Has made revisions in the manuscript.]
Comments 14: [Line 368 – Change “has” to “have,” since you are referring to multiple examples]
Response 14: [Has made revisions in the manuscript.]
Comments 15: [Line 378 – Given the ongoing ambiguity regarding aquatic insect identification, it would be helpful to include in the “Author Contributions” section a clear statement about which authors conducted the aquatic insect identification.]
Response 15: [Has made revisions in the manuscript.]
Comments 16: [Line 576 – A more descriptive title would be helpful here. Presumably, the list contains the aquatic macroinvertebrates you captured in your sampling. Therefore, “Macroinvertebrate taxa captured during sampling” is one suggestion for a more specific title. Also, if allowed by the journal, it would be helpful to more clearly separate the order name from the family and genus names. This could be done by bold print or underlining the order name. Also, you included some sub-families within the family column, so the heading should state “family or sub-family” to indicate this. There are also several mis-spellings in the taxonomic table including:
- For sp14 the correct family spelling is “Leptoceridae.”
- For sp26 the correct sub-family spelling is “Orthocladiinae.” Also, you listed the family name again under the genus column. Please correct this.
- For sp27 the sub-family is listed in the genus column too.
- For sp28 the sub-family is listed in the genus column too.
- For sp30 the stated order is undiscernible, but it looks like you intended to state “Deuterophlebiidae.” Also, there is only one genus “Deuterophlebia” for this family, so that spelling needs to be corrected too.
- For sp31 the family is listed in the genus column too.
- For sp47 you listed a genus in the family column. If the “Ephemera” genus here is correct, the family to which that genus belongs is “Ephemeridae.”]
Response 16: [Has made revisions in the manuscript.]
Comments 17: [Line 577-79 – The phrasing “the number of species of this species…” is awkward. Do you mean the number of captured specimens in this species? If so, please amend the phrasing to reflect this.]
Response 17: [We have revised the sentence as “Note: + means that the individuals of the species accounts for 0-1% of the total individuals captured; ++ means that the individuals of the species accounts for 1-10% of the total individuals captured; +++ means that the individuals of the species accounts for 10% of the total individuals captured. While, - indicates that the species is not distributed in the area (Value: Pollution resistance value, Up: Upstream, Mid: Midstream, Down: Downstream).”]